# Exogenous Melatonin Alleviates NaCl Injury by Influencing Stomatal Morphology, Photosynthetic Performance, and Antioxidant Balance in Maize

**DOI:** 10.3390/ijms251810077

**Published:** 2024-09-19

**Authors:** Fuqiang He, Xiaoqiang Zhao, Guoxiang Qi, Siqi Sun, Zhenzhen Shi, Yining Niu, Zefeng Wu, Wenqi Zhou

**Affiliations:** 1State Key Laboratory of Aridland Crop Science, College of Agronomy, Gansu Agricultural University, Lanzhou 730070, China; hefq6125@163.com (F.H.); qigx1321@163.com (G.Q.); 15045240973@163.com (S.S.); shizz@gsau.edu.cn (Z.S.); niuyn@gsau.edu.cn (Y.N.); wuzf@gsau.edu.cn (Z.W.); 2Crop Research Institute, Gansu Academy of Agricultural Sciences, Lanzhou 730070, China; zhouwenqi850202@163.com

**Keywords:** maize, melatonin, salt stress, reactive oxygen species, antioxidant system, stomata, photosynthesis, gene expression

## Abstract

Maize (*Zea mays* L.) is sensitive to salt stress, especially during seed germination and seedling morphogenesis, which limits maize growth and productivity formation. As a novel recognized plant hormone, melatonin (MT) participates in multiple growth and developmental processes and mediates biotic/abiotic stress responses, yet the effects of salt stress on maize seedlings remain unclear. Herein, we investigated the effects of 150 μM exogenous MT on multiple phenotypes and physiologic metabolisms in three-leaf seedlings across eight maize inbred lines under 180 mM NaCl salt stress, including growth parameters, stomatal morphology, photosynthetic metabolisms, antioxidant enzyme activities, and reactive oxygen species (ROS). Meanwhile, the six gene expression levels controlling antioxidant enzyme activities and photosynthetic pigment biosynthesis in two materials with contrasting salt resistance were examined for all treatments to explore the possible molecular mechanism of exogenous MT alleviating salt injury in maize. The results showed that 150 μM exogenous MT application protected membrane integrity and reduced ROS accumulation by activating the antioxidant system in leaves of maize seedlings under salt stress, their relative conductivity and H_2_O_2_ level average reduced by 20.91% and 17.22%, while the activities of superoxide dismutase (SOD), peroxidase (POD), catalase (CAT), and ascorbate peroxidase (APX) averaged increased by 13.90%, 17.02%, 22.00%, and 14.24% relative to salt stress alone. The improvement of stomatal size and the deposition of photosynthetic pigments were more favorable to enhancing photosynthesis in leaves when these seedlings treated with MT application under salt stress, their stomatal size, chlorophyll content, and net photosynthetic rate averaged increased by 11.60%, 19.64%, and 27.62%. Additionally, Gene expression analysis showed that MT stimulation significantly increased the expression of antioxidant enzyme genes (*Zm00001d009990*, *Zm00001d047479*, *Zm00001d014848*, and *Zm00001d007234*) and photosynthetic pigment biosynthesis genes (*Zm00001d011819* and *Zm00001d017766*) under salt stress. At the same time, 150 μM MT significantly promoted seedling growth and biomass accumulation. In conclusion, our study may unravel crucial evidence of the role of MT in maize seedlings against salt stress, which can provide a novel strategy for improving maize salt stress resistance.

## 1. Introduction

Soil salinity is one of the serious issues among all abiotic stresses; it severely limits agricultural production across the globe [1]. Salinity has devastatingly affected the area of more than 1.2 billion hectares [2], and high salinity is estimated to affect 20% and 33% of total cultivated and irrigated agricultural lands worldwide. By 2050, it is projected that over 50% of agricultural land will be salinized due to low precipitation, high surface evaporation, weathering of native rocks, irrigation with saline water, and poor cultural practices [3]. Soil salinity is a process of increasing the amount of soluble salt on the soil surface, which causes the loss of soil potential as a support for plant growth [4]. In cultivated land, salinity stress is commonly due to fertilization, poor quality of irrigation water, and land clearances [5]. Saline–alkali soils are not conducive to the growth and development of crops; they can damage crop tissues and result in a physiological drought [6]. Additionally, excessive salt will also cause the roots of crops to dehydrate and die [7].

Notably, most crops are salt sensitive. Maize (*Zea mays* L.) is an important crop for food, forage, and industrial purposes [8]. China, the second largest maize producer worldwide, contributes 23.5% of global maize production and 42.2% of total cereal production, which plays an important role in ensuring food security [9]. Maize is challenged by climate changes and abiotic stress factors such as soil salinization [10,11]. Fortunately, many chemicals, reagents, hormones, and growth regulators have been developed to deal with the harm of soil salinization to maize.

Melatonin (MT), an indole compound first identified in 1995 and found in both plants and animals, has been revealed to play important roles in the growth, development, and stress responses in plants [12,13,14]. Previous studies proved that MT could improve plant resistance against various environmental stresses, including salt, heavy metals, drought, and high/low temperatures [15,16,17]. In particular, foliar spraying of MT and root irrigation had significant alleviation effects on rice (*Oryza sativa* L.) resistance to salt stress [18]. MT promotes the production of salt-tolerant proteins, antioxidant enzymes, and defense-related molecules, thereby protecting the potato (*Solanum tuberosum* L.) from the harmful effects of abiotic stress [19]. Under salt stress, 70 μM MT treatments increase barley (*Hordeum vulgare* L.) plants’ photosynthetic efficiency, improve chlorophyll contents, reduce ROS (reactive oxygen species) generation, and thus alleviate oxidative damage to plants [20]. Under 150 mM NaCl stress, significant increases appeared in leaf area, biomass, and photosynthesis efficiency of *Brassica juncea* L. with MT treatments [21]. Thus, exogenous MT application could be an approach to alleviate stress injury in multiple plant species [22].

It is well known that maize possesses significant intraspecific genetic variations in salt resistance, and a few salt–resistant maize varieties have been developed [23]. However, salinity management for maize is still largely dependent on on-farm practices. This study focuses on the effect of exogenous MT on salt resistance in maize seedlings and its underlying mechanisms. Thus, the seedlings of eight elite maize inbred lines were treated with 150 μM exogenous MT application under 0 or 180 mM NaCl stress for seven days at the three-leaf stage, and the changes in growth phenotypes, ROS levels, membrane characteristics, antioxidant enzyme activities, stomatal morphology, pigments accumulation, and photosynthetic performance were measured. The gene expression levels of six candidate genes were further analyzed by real-time quantitative PCR (RT–qPCR). This will provide a valuable reference for improving salt resistance in maize, thus maintaining yield output in maize production in coping with the global salinization process.

## 2. Results

### 2.1. Effects of MT on the Growth of Maize Seedlings under Salt Stress

Under 180 mM NaCl + 0 μM MT treatment (SS), the seedling length (SL), seedling leaf area (SLA), seeding fresh weight (SFW), and root fresh weight (RFW) of eight maize genotypes were less than when they were treated with 0 mM NaCl + 0 μM MT treatment (CK); there were significant decreases with an average of 23.26%, 33.24%, 37.32, and 28.89%, respectively (Table 1). This indicates that NaCl stress had a negative effect on seedling growth and biomass accumulation in maize. These four traits were increased in 0 mM NaCl + 150 μM MT treatment (M–CK) compared to CK by 10.08%, 16.60%, 16.73%, and 18.10%, respectively (Table 1). At the same time, they have increased in 180 mM NaCl + 150 μM MT treatment (M–SS) compared to SS treatment significantly by 17.01%, 28.61%, 32.32%, and 25.01%, respectively (Table 1). This suggests that MT application could significantly alleviate salt injury of maize seedlings to maintain overall health and growth potential.

### 2.2. Effects of MT on Photosynthetic Parameters of Maize Seedlings under Salt Stress

Photosynthetic parameters can reflect the physiological status and changes in maize seedlings under salt stress [24]. In this study, the net photosynthetic rate (Pn) of leaves in all maize seedlings was significantly reduced by 18.79~58.27% in SS treatment compared to CK, the stomatal conductance (Gs) was significantly reduced by 21.13~56.80%, intercellular CO_2_ concentration (Ci) was significantly increased by 21.74~132%, and the transpiration rate (Tr) was significantly reduced by 17.75~40.19% (Figure 1). Unlikely, compared to the seedlings in CK treatment, when they were exposed to M–CK treatment, the changes in Pn in NX32–1 and XQ4–1 seedlings were not obvious. The average Pn of the other six materials increased significantly by 22.01%; the changes in Gs in LX19 and GZ13–2 seedlings were not obvious; the average Gs of other maize significantly increased by 17.83%. The increase in Tr in XQ4-1 seedling was slight; the average Tr of other genotypes significantly increased by 22.35% (Figure 1). Moreover, compared to SS treatment, the Pn, Gs, and Tr averages were significantly increased by 27.62%, 35.73%, and 16.09% in M–SS-treated seedlings, respectively, while the average Ci was significantly reduced by 20.29% (Figure 1).

### 2.3. Effects of MT on Chlorophyll Accumulation of Maize Seedlings under Salt Stress

For the leaves of eight maize seedlings genotypes, their chlorophyll a (Chl a) content under SS treatment significantly reduced by 5.58% to 36.17% compared to CK; the chlorophyll b (Chl b) content in leaves of XY201 and ZC2–7 seedlings showed a slight decrease, that in other maize materials significant reduced by 6.68% to 23.14%; the chlorophyll a/b (Chl a/b) in the leaves of LX19 and NX32–1 seedlings showed a slight decrease, that in other seedlings significant reduced by 11.81% to 21.10%; the carotenoid (Car) content in the leaves of LX19 seedling showed a significant increase, that in GZ13–2 seedling displayed a slight decrease, while the Car content in other maize genotypes significant reduced by 16.79% to 53.24%; the SPAD value in the leaves of LX19 and GZ13–2 seedlings showed a slight decrease, that in other maize genotypes significantly reduced 11.86% to 16.73% (Figure 2). Moreover, for the eight maize seedlings, compared to CK, M–CK treatment increased the average Chl a content by 16.14%, average Chl b content by 9.97%, average Car content by 23.83%, and average SPAD significantly by 7.63% (Figure 2). Additionally, M–SS treatment significantly increased the Chl a content by 26.48%, Chl b content by 13.44%, Car content by 27.61%, and SPAD content by 6.54% compared to the SS treatment (Figure 2). The results showed that 180 mM NaCl stress could inhibit Chl a, Chl b, and Car production and decrease SPAD value in maize. The application of 150 μM exogenous MT could promote chlorophyll formation, enhance light absorption and transmission, maintain higher photosynthesis during maize seedling development under salt stress, and lastly, effectively reduce the damage to maize.

### 2.4. Effects of MT on Antioxidant Enzyme Activities and H_2_O_2_ Content of Maize Seedlings under Salt Stress

For the leaves of all maize seedlings under SS treatment, the superoxide dismutase (SOD) activity significantly increased by 16.96% to 82.03% compared to CK, except for GS36–2 seedling, the peroxidase (POD) activity significantly increased by 18.30% to 106.00%, the catalase (CAT) activity significantly increased by 48.97% to 163%, and the ascorbate peroxidase (APX) activity significantly increased by 17.86% to 42.75% (Figure 3). Moreover, M–CK-treated leaves of the eight maize materials showed a 20.15% increase in the average SOD activity, a 22.96% increase in average POD activity, a 36.26% increase in average CAT activity, and a 13.87% increase in average APX activity compared to CK treatment (Figure 3). Like the M–CK treatment, all seedlings were cultured in the M–SS treatment and the activities of the four antioxidant enzymes had significant increases of 13.90%, 17.02%, 22.01%, and 14.24%, respectively (Figure 3). In addition, the H_2_O_2_ content of all maize seedlings displayed a significant increase (ranging from 21.26% to 62.62%) under SS treatment relative to CK-treated seedlings, while it had a significant decrease under M–CK (27.96~52.77%) and M–SS (8.85~24.49%) treatments compared to SS treatment (Figure 3).

### 2.5. Effects of MT on Stomatal Morphology of Maize Seedlings under Salt Stress

There were no significant differences in the stomatal length of the adaxial surface (SL–Z) and stomatal length of the abaxial surface (SL–F) of leaves in the eight maize seedlings under both SS and M–CK treatments compared to CK (Figure 4A,B). SS treatment significantly reduced the stomatal width of the adaxial surface (SW–Z) and stomatal width of the abaxial surface (SW–F) of these maizes’ leaves compared to CK (Figure 4C,D), their SW–Z clearly reduced by 7.50~18.71% (Figure 4C) and their SW–F clearly reduced by 10.52~24.37% (Figure 4D). Under the M–CK treatment, the SW–Z and SW–F of all maizes were significantly higher than those under CK, with a significant increase of 9.41~21.26% in SW–Z (Figure 4C) and 7.50~16.83% in SW–F (Figure 4D). M–SS treatment significantly increased the SW–Z and SW–F compared to SS treatment, with a significant increase in SW–Z ranging from 7.89% to 22.78% (Figure 4C). The SW–F of XY201 leaves was slightly increased, and the average SW–F of seven seedlings significantly increased by 10.45%. Compared with CK, the stomatal area of the adaxial surface (SA–Z) and stomatal area of the abaxial surface (SA–F) in SS-treated maize seedlings also significantly reduced (Figure 4E,F), with the SA–Z significantly reduced by 12.03~25.64% (Figure 4E), and the SA–F significantly reduced by 11.76~26.12% (Figure 4F). The SA–Z and SA–F of all maize seedlings under the M–CK treatment were higher than those under CK, which showed a 14.64% and an 18.86% increase, respectively (Figure 4E,F). Additionally, compared to the SS treatment, the SA–Z had a 10.02~21.10% increase, and the SA–F had a 6.90~13.00% increase (except for the NX40–6 seedling) under M–SS treatment (Figure 4E,F).

### 2.6. Effects of MT on Relative Water Content and Membrane Characteristics of Maize Seedlings under Salt Stress

Compared to CK, there was lower relative water content (RWC) in leaves of all maize materials under SS treatment (6.91~10.82% decrease), while M–CK treatment had no effect on RWC (Figure 5A). Compared to SS treatment, however, M–SS treatment caused a significant increase in RWC content by 5.98% to 13.05% (Figure 5A). Moreover, SS treatment aggravated the plasma membrane permeability in maize, resulting in a significant increase in the relative conductivity (REC; 13.59~119%) compared to CK (Figure 5B). An interesting phenomenon was also observed: the REC decreased in eight maize genotype seedlings under M–CK treatment compared to when they were under CK. At the same time, it showed a 20.91% decrease in leaves of eight maize genotypes under M–SS treatment relative to SS stress (Figure 5B).

### 2.7. Expression Analysis of Six Candidate Genes

To reveal the expression levels of antioxidant enzyme genes and photosynthetic pigments biosynthetic genes to response salt tolerance in maize under different treatments. Therefore, the four antioxidant enzyme genes, including *Zm00001d009990* (encoding superoxide dismutase [Mn] 3.4, mitochondrial), *Zm00001d047479* (encoding superoxide dismutase [Cu-Zn] 4AP), *Zm00001d014848* (encoding catalase 1), *Zm00001d007234* (encoding ascorbate peroxidase 2), and two photosynthetic pigments biosynthetic genes, i.e., *Zm00001d011819* (a chlorophyllide-a-oxygenase chloroplastic) and *Zm00001d017766* (a nine–cis–epoxycarotenoid dioxygenase 8) were randomly selected to perform gene qRT-PCR analysis across LX19 and NX40-6 leaves under four treatments. The results showed that the expression levels of four antioxidant enzyme genes significantly enhanced in LX19 and NX40-6 leaves under SS stress compared to CK treatment (*p* < 0.05) (Figure 6), while the expression of two photosynthetic pigment biosynthesis genes significantly reduced with 23.59~31.25% and 30.23~31.48% (*p* < 0.05) (Figure 6); Further the M–SS treatment induced upregulation of all genes expression levels, with 2.93~20.81% and 5.47~35.78% compared to SS stress (Figure 6). It is speculated that the activated or inhibited expression of these candidate genes regulated various antioxidant enzyme activities and photosynthetic pigment formation. These changes in physiological metabolisms could influence salt tolerance in maize under SS stress and MT stimulation.

### 2.8. Correlation Relationships among Traits and Candidate Genes of Maize Seedlings under All Treatments

To further understand the complex relationship networks of 26 traits and six candidate gene expressions, their Pearson correlation analysis was performed across LX19 and NX40-6 leaves under four treatments (Figure 7). Interestingly, there were 207 groups with significant (*p* < 0.01 or *p* < 0.05) correlations between both traits, 15 groups with significant (*p* < 0.01 or *p* < 0.05) correlations between both genes, as well as 119 groups with significant (*p* < 0.01 or *p* < 0.05) correlations between trait and gene (Figure 7). Such as, the expression levels of *Zm00001d047479* (Cu-Zn SOD) showed significantly positive correlation to REC, Ci, SOD activity, POD activity, CAT activity, APX activity, and H_2_O_2_ level, and showed significantly negative correlation to SL, SLA, SFW, RFW, RWC, Pn, Gs, Tr, SL–F, SW–F, SA–F, SW–Z, Chl a content, Chl b content, Chl a/b, and SPAD value. The expression levels of photosynthetic pigments biosynthesis-related gene, i.e., *Zm00001d017766*, showed a significantly positive correlation to SL, RFW, Pn, Gs, Tr, SW–F, SA–F, SW–Z, Chl a content, Chl b content, Car content, Chla/b, and SPAD value, and showed significantly negative correlation to REC, Ci, SOD activity, CAT activity, APX activity, and H_2_O_2_ content (Figure 7). These findings thus showed that the six candidate genes for antioxidant enzyme activity and photosynthetic pigment biosynthesis could cooperate with each other to directly or indirectly regulate multiple metabolism and development processes, including seedling growth, stomatal morphology, photosynthetic performance, antioxidant system, ROS homeostasis, and membrane integrity in maize under diverse environments.

## 3. Discussion

It is well known that salt stress markedly inhibits plant growth and causes crop yield loss [25]. Fortunately, various plants have developed different strategies to respond to environmental stresses. Exogenous MT application significantly enhanced plant resistance to adverse environmental conditions [26]. In the present experiment, the positive protective roles of MT in maize seedlings against salt stress were deeply investigated.

Previous works indicated that MT might play an important regulatory role in plant growth and development [27,28,29]. In this study, we found that the growth of maize was notably inhibited when exposed to salt stress. The average SL, SLA, SFW, and RFW of the eight maize materials showed significant reductions of 23.26%, 33.24%, 37.32%, and 28.89%, respectively (Table 1). Furthermore, 150 μM exogenous MT application significantly improved maize seedlings’ growth under 180 mM salt stress. These growth parameters were improved, with 17.01%, 28.61%, 32.32%, and 25.01% significant increases in SL, SLA, SFW, and RFW, respectively (Table 1). This was consistent with previous studies [29,30,31].

At the same time, early studies also showed that MT might inhibit the decline in photosynthetic pigments under different environmental stresses [29,32]. Our results showed that 150 μM exogenous MT application caused a significant increase in contents of Chl a (19.64%), Chl b (13.44%), and Car (27.61%) in maize seedlings under 180 mM salt stress (Figure 2A,B,D). Meanwhile, the expression of the photosynthetic pigment biosynthesis genes, i.e., *Zm00001d011819* and *Zm00001d017766*, significantly enhanced when treated with 150 μM MT application under 180 mM salt stress, implying that these genes positively controlled photosynthetic pigments biosynthesis under MT stimulation, subsequently increasing the absorption and utilization of light energy to enhance photosynthesis in maize. The SPAD value was also significantly higher in the M–SS treatment than the SS treatment (Figure 2E). We also found that under 180 mM salt stress, the 150 μM exogenous MT application caused a significant increase of 12.87% in SW–Z, 10.45% in SW–F, 15.45% in SA–Z, and 7.74% in SA–F of maize seedlings (Figure 4C–F). A previous study found that the application of MT treatment has been shown to increase Gs [33]. This was consistent with our study. Therefore, exogenous MT may induce photosynthetic pigment biosynthesis, improve the photosystem, activate antioxidant enzyme activities, and regulate stomatal morphology to maintain the normal photosynthetic process. As a result, the Pn increased by 27.62%, and Gs increased by 35.73% in the leaves of all maize seedlings that were treated with 150 μM exogenous MT application under 180 mM NaCl stress (Figure 1A,B).

It has been known that abiotic stress usually causes a clear decrease in leaf water status [34,35]. Similarly, we observed that salt stress led to a significant decline in RWC (approximately a decrease of 8.81%) (Figure 5A). In addition, salt stress can induce osmotic stress and ionic toxicity, which in turn can lead to the production of excess ROS and oxidative stress [36,37]. To reduce stress–triggered ROS accumulation, plants have evolved an effective antioxidant system, including enzymatic and non-enzymatic antioxidants. In plants, MT has also been suggested to be a crucial antioxidant that can scavenge oxygen free radicals effectively [28]. Many studies have indicated that exogenous MT application can increase some antioxidant enzyme (such as POD, SOD, and APX) activities under abiotic stress in plants [19,34,38]. In this experiment, we also found that the activities of SOD, POD, CAT, and APX were higher in the presence of MT under salt stress, and the expression of antioxidant enzyme genes (*Zm00001d009990*, *Zm00001d047479, Zm00001d014848*, and *Zm00001d007234*) were significantly enhanced, which may be mainly due to the synergistic effect of salt stress and exogenous MT. Therefore, the positive effects of exogenous MT on the active oxygen scavenging system could improve the stress resistance of maize seedlings. Exogenous MT significantly suppressed the production of hydroxyl radicals and H_2_O_2_ [19]. Consistently, our research showed that salt stress caused a 40.98% increase in H_2_O_2_ accumulation in maize leaves, but 150 μM exogenous MT obviously alleviated H_2_O_2_ accumulation (Figure 3E). Appropriate concentrations of ROS are necessary signaling molecules in plants. On the one hand, excessive ROS can cause lipid peroxidation membrane disruption; on the other hand, excessive ROS can also lead to enzyme inactivation and metabolic abnormalities [38,39,40,41]. In this present study, H_2_O_2_ levels of all seedlings were significantly lower in the M–SS treatment than in the SS treatment (Figure 3E), and changes in REC were consistent (Figure 5B), further indicating that MT might protect cell membranes against oxidative damage induced by salt stress [34]. Interestingly, MT level may be associated with antioxidant ability in response to environmental stresses in plants [42]. In the present study, under 180 mM salt stress condition, the addition of 150 μM exogenous MT significantly increased the activities of antioxidant enzymes, i.e., SOD activity increased by 13.90%, POD activity increased by 17.02%, CAT activity increased by 22.00%, and APX activity increased by 14.24% in maize seedlings (Figure 3).

Taken together, our data provide detailed protective evidence for exogenous MT in maize under salt stress by investigating the stomatal morphology, antioxidative defense system, and photosynthetic machinery. This protective role may be involved in the activation of the antioxidant defense system, the elimination of ROS, and the protection of photosynthetic apparatus. For this purpose, we try to construct a possible mechanistic map of the role of MT in the improvement of salt resistance in maize seedlings under NaCl stress (Figure 8), which will provide a basis for a systematic understanding of molecular mechanisms on exogenous MT alleviating salt injury in maize.

## 4. Materials and Methods

### 4.1. Maize Materials and Treatments

The random eight elite maize genotypes, including XY201, LX19, GS36–2, ZC2–7, NX32–1, NX40–6, GZ13–2, and XQ4–1 from Molecular Biology Laboratory, Stata Key Laboratory of Aridland Crop Science, China, were used in this study. The seeds of eight genotypes were separately sterilized with 0.5% (*v*/*v*) sodium hypochlorite solution for 15 min, rinsed five times with double–distilled water (ddH_2_O), and soaked in ddH_2_O for 24 h at 22 ± 0.5 °C indoor environment. Then, the ten soaked seeds were planted in plastic boxes and cultured in a greenhouse (25 ± 0.5 °C constant temperature; 300 μM m^−2^ s^−1^ light intensity; 65% relative humidity) for 15 d. The three–leaf seedlings were then treated with four treatments for 7 d, i.e., 0 mM NaCl + 0 μM MT (ddH_2_O, CK), 180 mM NaCl + 0 μM MT(SS), 0 mM NaCl + 150 μM MT(M–CK), and 180 mM NaCl + 150 μM MT(M–SS). During all treatments, the seedlings were replenished with 50 mL of the corresponding mixed solution every two days. Each of the above treatments was done in three replicates.

### 4.2. Growth Performances Observation

For the seedlings of the above eight maize genotypes under four treatments, their seedling length (SL), seedling fresh weight (SFW), and root fresh weight (RFW) were measured according to the Zhao et al. [43] method. The seedling leaf area (SLA) was calculated as follows:SLA = leaf length × leaf width × 0.75(1)

### 4.3. Determination of Chlorophyll and Photosynthetic Parameters

The relative chlorophyll content (SPAD) [44] value of the third leaf of maize seedlings under all treatments was determined using the chlorophyll meter (SPAD-502; Konica Minolta Sensing, Inc., Japan). Four photosynthetic parameters of the third leaf of maize seedlings under these treatments were measured between 10:30 and 11:30 AM using the LI-6400 XT portable photosynthesis system (LI-COR; Biosciences, Inc. Lincoln, NE, USA), including net photosynthetic rate (Pn; μM CO_2_·m^−2^·s^−1^), intercellular CO_2_ (Ci; μM CO_2_·mol^−1^), stomatal conductance (Gs; M H_2_O·m^−2^·s^−1^), and transpiration rate (Tr; mM H_2_O·m^−2^·s^−1^), they were measured in a chamber at 1500 μM m^−2^·s^−1^ photosynthetically active radiation and 380 ± 5 μM CO_2_/M.

The 1.0 g fresh leaves sample from each treatment was placed in 10 mL of 95% alcohol (*v*/*v*) for 48 h in darkness. The absorbance values of extracting solution were then measured using a multifunctional enzyme marker model (SynergyHTX; BioTek Instruments, Inc. USA) at 665 nm, 649 nm, and 470 nm. The concentrations of chlorophyll a (Ca), chlorophyll b (Cb), and carotenoids (Cc) were calculated as follows:Ca = 13.95 × A665 − 6.88 × A649 (2)
Cb = 24.96 × A649 − 7.32 × A665 (3)
Cc = 1000 × A470 − 2.05 × Ca – 114 × Cb (4)

The contents of these pigments were then calculated as follows:chlorophyll a content (mg/g) = Ca × Vt × n/FW × 1000(5)
chlorophyll b content (mg/g) = Cb × Vt × n/FW × 1000 (6)
carotenoid content (mg/g) = Cc × Vt × n/FW × 1000 (7)
where Ca, Cb, and Cc represented the concentrations of chlorophyll a (Chl a), chlorophyll b (Chl b), and carotenoids (Car), respectively. FW referred to the fresh weight of the sample (g), Vt was the total volume of extracting solution (mL), and n was the dilution factor of extracting solution.

### 4.4. Determination of Antioxidant Enzyme Activity and H_2_O_2_ Content

Refer to the method of Zhao et al. [43], the SOD activity, POD activity, CAT activity, APX activity, and H_2_O_2_ content were determined using the corresponding Solarbio kits (Beijing Solarbio Science and Technology Co., Ltd., Beijing, China) and the multi-function microplate reader (SynergyHTX; BioTek Instruments, Inc., South Burlington, VT, USA), following the manufacturer’s kit instructions.

### 4.5. Characterization of Stomatal Morphology

The third fresh leaf of maize seedlings under each treatment was cut, and the middle of the leaf (1 cm ×1 cm) was stuck on the slide coated with glue. Peeling off the adaxial and abaxial surfaces after 20 min. Stomatal morphology was observed using a forward and inverted integrated fluorescence microscope (Revolve RVL–100–G, ECHO, Lake Zurich, IL, USA). Five fields of view were randomly selected for each treatment on the front and back sides. Stomatal length, width, and area were measured and statistically recorded under a 4× objective (image size 3226 × 3024).

### 4.6. Determination of Relative Water Content and Relative Conductivity

The 1.0 g fresh leaves (FW) of the seedlings under each treatment were measured and then completely immersed in 30 mL ddH_2_O for 12 h until constant water (TW). The leaves were then dried in an oven until the constant weight (DW). The relative water content (RWC) was calculated as follows:[(FW − DW)/(TW − DW)] × 100%.(8)

The 0.1 g fresh leaves of the seedlings under each treatment were placed into the test tubes containing 10 mL ddH_2_O and soaked at room temperature for 12 h. The conductivity (R_1_) was measured by a DDSJ-308F conductivity meter (Rex Electric Chemical, Shanghai, China). Then, the same set of samples was stored in a 100 °C water bath for 15 min, and the electrical conductivity (R_2_) was recorded. The REC was estimated as follows [40]:REC = (R_1_/R_2_) × 100%.(9)

### 4.7. Data Statistical Analyses

All data were shown as means ± SE (standard error). At least three independent replicates were conducted for each treatment. Statistical analysis was carried out using SPSS Statistics software (V.21.0, SPSS, Chicago, IL, USA). Significant differences were determined using one-way ANOVA and Duncan test at *p* < 0.05 level. Pearson correlation analysis was performed using Origin 2022 (Version 22.0, OriginPro, Northampton, MA, USA).

### 4.8. RT-qPCR Analysis

According to the changes of phenotypes and physiological metabolisms of eight maize genotypes under all treatments, we selected LX19 and NX40–6 seedlings, with contrasting salt tolerance to extract their total RNAs with TRIZOL reagent (Invitrogen, Waltham, MA, USA), which was then reverse–transcribed into cDNA using a SuperScript III First-Strand Kit (Invitrogen). The qRT-PCR was conducted using TransStart Tip Green qPCR SuperMix (Tran, Beijing, China). Primers (Appendix A) for these candidate genes for antioxidant enzymes and photosynthetic pigment biosynthesis were designed via the Primer3web v.4.1.0 (https://primer3.ut.ee/; accessed on 1 May 2024). The positions of these genes were mapped in the *Zea_mays* B73_V4 reference genome (https://www.maizegdb.org/; accessed on 12 May 2024), and their functional annotation was performed using the tool AgBase v.2.00 (https://agbase.arizona.edu/; accessed on 18 May 2024). Relative gene expression levels were calculated by the 2^−∆∆Ct^ method, with *Zm00001d010159* as an internal reference gene.

## 5. Conclusions

In conclusion, 150 μM exogenous MT application could enhance the NaCl resistance of maize seedlings through the following pathways: (1) MT-activated photosynthetic pigment biosynthesis genes expression (*Zm00001d011819*, *Zm00001d017766*), prevented chlorophyll degradation, and improved stomata morphology, thus improving photosynthetic capacity under NaCl stress. (2) MT up-regulated the expression of antioxidant enzyme genes (*Zm00001d009990*, *Zm00001d047479, Zm00001d014848*, *Zm00001d007234*), which enhanced the activities of antioxidant enzymes, reduced excessive accumulation of ROS, and inhibited membrane lipid peroxidation. (3) These candidate genes, growth phenotypes, and physiological metabolisms interacted with each other to form a complex regulatory mechanism to respond to salt tolerance under NaCl stress and MT stimulation. Overall, the application of exogenous MT has the potential to improve maize seedlings’ growth under NaCl stress.

## Figures and Tables

**Figure 1 ijms-25-10077-f001:**
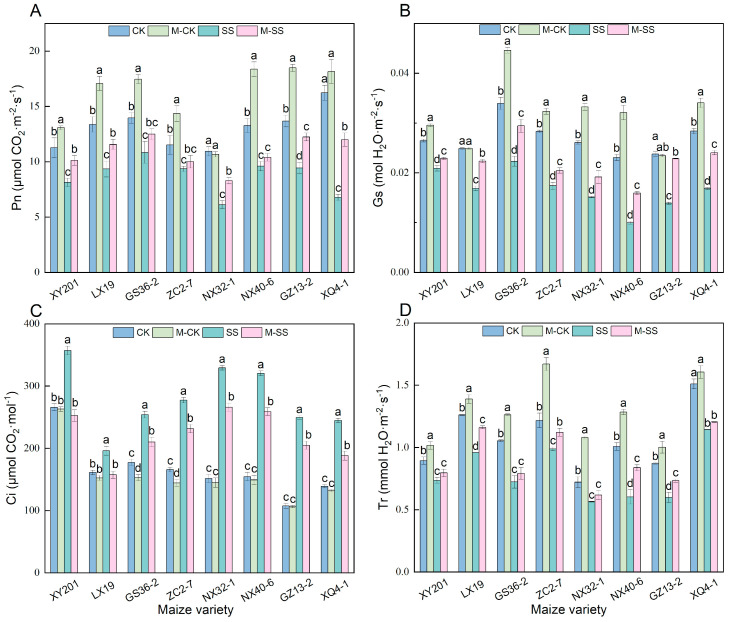
Changes in photosynthetic parameters of all maize seedlings under different treatments. Different lowercase letters indicated significant differences in *p* < 0.05 level. (**A**): net photosynthetic rate (Pn); (**B**): stomatal conductance (Gs); (**C**): intercellular CO_2_ concentration (Ci); (**D**): transpiration rate (Tr). CK: 0 mM NaCl + 0 μM melatonin (MT) treatment; SS: 180 mM NaCl + 0 μM MT treatment; M–CK: 0 mM NaCl + 150 μM MT treatment; M–SS: 180 mM NaCl + 150 μM MT treatment.

**Figure 2 ijms-25-10077-f002:**
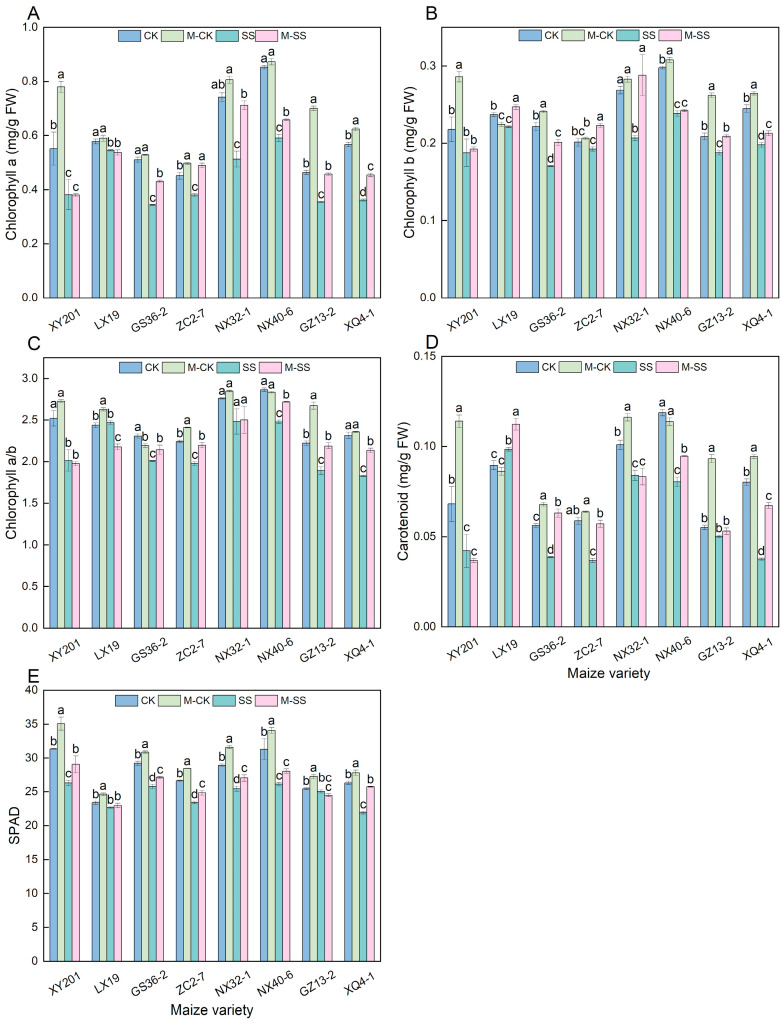
Changes in chlorophyll of maize seedlings under different treatments. Different lowercase letters indicated significant differences in *p* < 0.05 level. (**A**): chlorophyll a content (Chl a); (**B**): chlorophyll b content (Chl b); (**C**): chlorophyll a content/chlorophyll b content (Chl a/b); (**D**): carotenoid content (Car); (**E**): relative chlorophyll content (SPAD). CK: 0 mM NaCl + 0 μM melatonin (MT) treatment; SS: 180 mM NaCl + 0 μM MT treatment; M–CK: 0 mM NaCl + 150 μM MT treatment; M–SS: 180 mM NaCl + 150 μM MT treatment.

**Figure 3 ijms-25-10077-f003:**
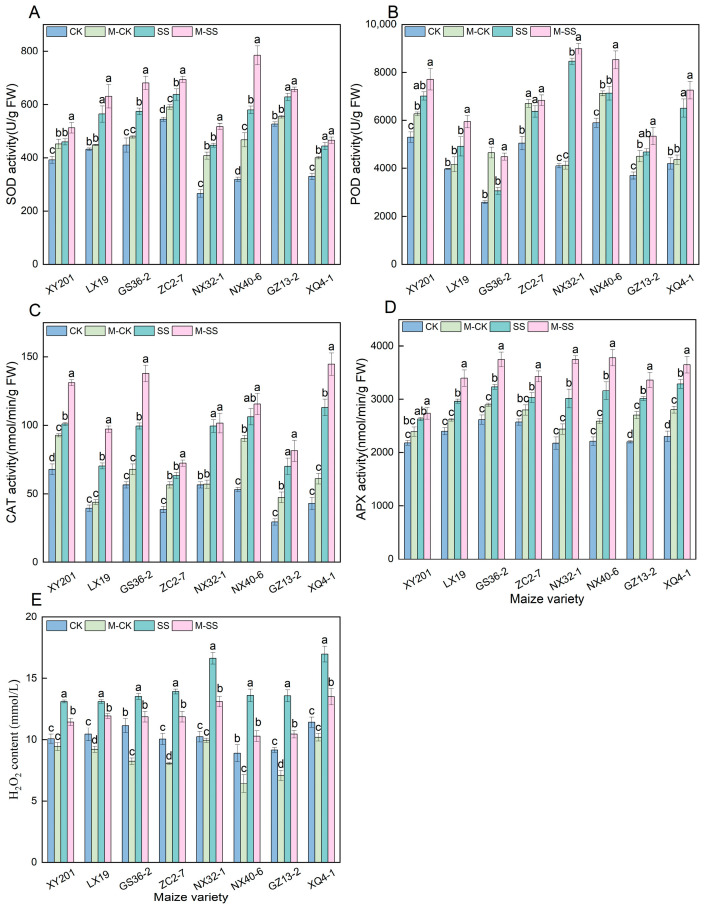
Changes in Antioxidant Enzyme Activities and H_2_O_2_ Content of all maize seedlings under different treatments. Different lowercase letters indicated significant differences in *p* < 0.05 level. (**A**): superoxide dismutase (SOD); (**B**): peroxidase (POD); (**C**): catalase (CAT); (**D**): ascorbate peroxidase (APX); (**E**): hydrogen peroxide (H_2_O_2_). CK: 0 mM NaCl + 0 μM melatonin (MT) treatment; SS: 180 mM NaCl + 0 μM MT treatment; M–CK: 0 mM NaCl + 150 μM MT treatment; M–SS: 180 mM NaCl + 150 μM MT treatment.

**Figure 4 ijms-25-10077-f004:**
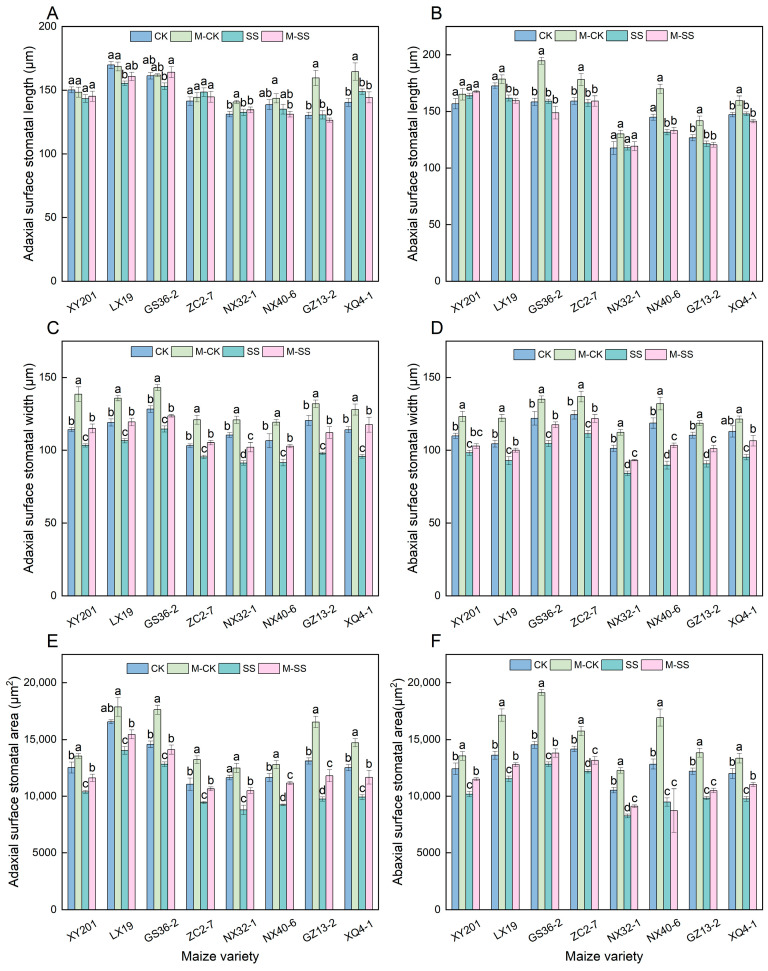
Changes in stomatal morphology of all maize seedlings under different treatments. Different lowercase letters indicated significant differences in *p* < 0.05 level. (**A**): stomatal length of adaxial surface (SL–Z); (**B**): stomatal length of abaxial surface (SL–F); (**C**): stomatal width of adaxial surface (SW–Z); (**D**): stomatal width of abaxial surface (SW–F); (**E**): stomatal area of adaxial surface (SA–Z); (**F**): stomatal area of abaxial surface (SA–F). CK: 0 mM NaCl + 0 μM melatonin (MT) treatment; SS: 180 mM NaCl + 0 μM MT treatment; M–CK: 0 mM NaCl + 150 μM MT treatment; M–SS: 180 mM NaCl + 150 μM MT treatment.

**Figure 5 ijms-25-10077-f005:**
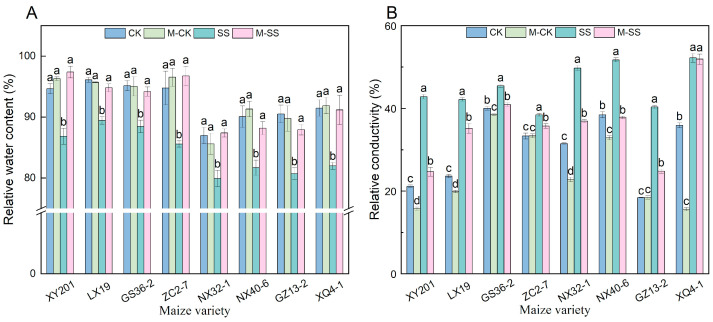
Changes in relative water content (RWC; (**A**)) and relative conductivity (REC; (**B**)) of all maize seedlings under different treatments. Different lowercase letters indicated significant differences in *p* < 0.05 level. CK: 0 mM NaCl + 0 μM melatonin (MT) treatment; SS: 180 mM NaCl + 0 μM MT treatment; M–CK: 0 mM NaCl + 150 μM MT treatment; M–SS: 180 mM NaCl + 150 μM MT treatment.

**Figure 6 ijms-25-10077-f006:**
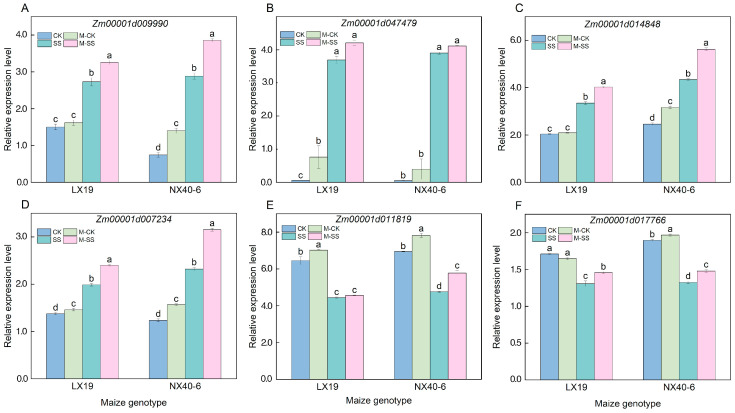
Relative expression levels of six random candidate genes across LX19 and NX40-6 maize varieties under four treatment conditions, including superoxide dismutase [Mn] 3.4, mitochondrial (*Zm00001d009990* (**A**)), superoxide dismutase [Cu-Zn] 4AP (*Zm00001d047479* (**B**)), catalase 1 (*Zm00001d014848* (**C**)), ascorbate peroxidase 2 (*Zm00001d007234* (**D**)), a chlorophyllide-a-oxygenase chloroplastic (*Zm00001d011819* (**E**)), a nine–cis–epoxycarotenoid dioxygenase 8 (*Zm00001d017766*) (**F**).The different letters indicated significant differences between the two treatments within a maize genotype (*p* < 0.05). Different lowercase letters indicated significant differences in *p* < 0.05 level. CK: 0 mM NaCl + 0 μM melatonin (MT) treatment; SS: 180 mM NaCl + 0 μM MT treatment; M–CK: 0 mM NaCl + 150 μM MT treatment; M–SS: 180 mM NaCl + 150 μM MT treatment.

**Figure 7 ijms-25-10077-f007:**
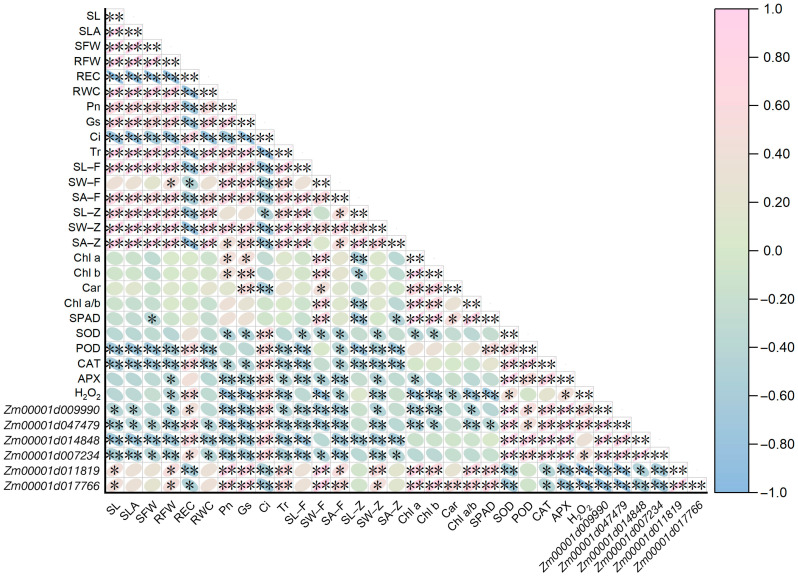
Pearson correlation analysis among 26 traits and six candidate genes across LX19 and NX40–6 seedlings under four treatments. SL: seedling length; SLA: seedling leaf area; SFW: seedling fresh weight; RFW: root fresh weigh; REC: relative conductivity; RWC: relative water content; Pn: net photosynthetic rate; Gs: stomatal conductance; Ci: intercellular CO_2_ concentration; Tr; transpiration rate; SL–Z: stomatal length of adaxial surface; SW–Z: stomatal width of adaxial surface; SA–Z: stomatal area of adaxial surface; SL–F: stomatal length of abaxial surface; SW–F: stomatal width of abaxial surface; SA–F: stomatal area of abaxial surface; Chl a: chlorophyll a content; Chl b: chlorophyll b content; Car: carotenoid content; Chl a/b: chlorophyll a content/chlorophyll b content; SPAD: SPAD content; SOD: superoxide dismutase activity; POD: peroxidase activity; CAT: catalase activity; APX: ascorbate peroxidase activity; H_2_O_2_: H_2_O_2_ content. ** indicated significant correlations in *p* < 0.01 level. while * indicated significant correlations in *p* < 0.05 level. The graph displays a red ball to represent a positive correlation between the two indicators, and a blue ball to represent a negative correlation. The strength of the correlation is indicated by the color and volume of the ball.

**Figure 8 ijms-25-10077-f008:**
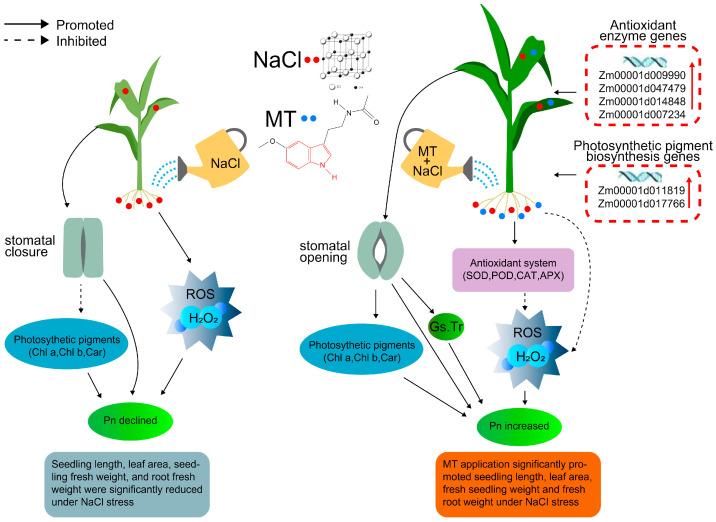
Schematic model of response mechanism for exogenous melatonin (MT) alleviates salt injury in maize seedlings. Solid arrows indicated promote effects, dashed arrows indicated inhibited effects, red arrows indicated positive expression of genes; ROS: reactive oxygen species; SOD: superoxide dismutase; POD: peroxidase, CAT: catalase; APX: ascorbate peroxidase; Pn; net photosynthetic rate; Gs: stomatal conductance; Tr: transpiration rate.

**Table 1 ijms-25-10077-t001:** Changes in phenotypes observations in all maize seedlings under different treatments.

Maize Variety	Treatment	SL(cm)	SLA (cm^2^)	SFW (g)	RFW (g)
XY201	CK	20.47 ± 0.22 ab	17.78 ± 0.19 ab	1.13 ± 0.05 b	1.57 ± 0.07 a
	M–CK	22.23 ± 0.79 a	19.26 ± 0.86 a	1.33 ± 0.04 a	1.69 ± 0.11 a
	SS	16.13 ± 0.58 c	13.06 ± 0.28 c	0.80 ± 0.03 c	1.27 ± 0.05 b
	M–SS	19.37 ± 0.43 b	16.90 ± 0.22 b	1.13 ± 0.06 b	1.46 ± 0.09 ab
LX19	CK	26.80 ± 0.42 b	21.07 ± 0.96 b	1.52 ± 0.06 b	1.86 ± 0.18 b
	M–CK	29.97 ± 0.96 a	24.58 ± 0.46 a	1.89 ± 0.09 a	2.46 ± 0.19 a
	SS	20.83 ± 0.52 c	15.84 ± 0.13 c	1.03 ± 0.13 c	1.35 ± 0.07 b
	M–SS	25.30 ± 0.57 b	20.95 ± 0.29 b	1.26 ± 0.06 bc	1.69 ± 0.15 b
GS36-2	CK	24.07 ± 0.85 b	19.36 ± 0.55 b	1.29 ± 0.02 b	1.38 ± 0.05 b
	M–CK	26.60 ± 0.87 a	25.32 ± 0.77 a	1.68 ± 0.05 a	1.86 ± 0.11 a
	SS	17.63 ± 0.75 d	12.65 ± 0.17 d	0.85 ± 0.04 d	1.06 ± 0.13 c
	M–SS	20.67 ± 0.39 c	15.76 ± 0.67 c	1.08 ± 0.06 c	1.32 ± 0.05 bc
ZC2-7	CK	24.70 ± 0.49 ab	19.58 ± 0.97 b	1.47 ± 0.07 a	2.01 ± 0.09 a
	M–CK	26.40 ± 0.61 a	22.89 ± 1.22 a	1.48 ± 0.03 a	2.05 ± 0.04 a
	SS	20.26 ± 1.16 c	14.14 ± 0.35 c	0.95 ± 0.01 b	1.49 ± 0.09 b
	M–SS	22.73 ± 0.74 bc	16.96 ± 0.93 bc	1.10 ± 0.11 b	1.67 ± 0.09 b
NX32-1	CK	18.30 ± 0.61 a	13.53 ± 0.28 b	0.71 ± 0.03 a	1.11 ± 0.04 b
	M–CK	19.66 ± 0.39 a	15.51 ± 0.27 a	0.76 ± 0.02 a	1.30 ± 0.05 a
	SS	14.20 ± 0.42 c	8.23 ± 0.28 d	0.44 ± 0.01 c	0.81 ± 0.04 c
	M–SS	16.50 ± 0.62 b	11.17 ± 0.71 c	0.56 ± 0.06 b	0.97 ± 0.06 b
NX40-6	CK	21.70 ± 0.58 b	14.63 ± 0.42 b	0.86 ± 0.02 ab	1.51 ± 0.15 ab
	M–CK	23.76 ± 0.85 a	18.29 ± 0.75 a	0.98 ± 0.08 a	1.78 ± 0.13 a
	SS	16.10 ± 0.25 d	8.25 ± 0.19 d	0.51 ± 0.05 c	0.81 ± 0.11 c
	M–SS	19.43 ± 0.49 c	10.62 ± 0.36 c	0.77 ± 0.06 b	1.19 ± 0.08 bc
GZ13-2	CK	23.63 ± 0.64 b	19.20 ± 0.28 b	1.06 ± 0.06 b	1.98 ± 0.17 ab
	M–CK	27.50 ± 0.95 a	20.57 ± 0.58 a	1.35 ± 0.06 a	2.26 ± 0.14 a
	SS	17.23 ± 0.20 c	12.37 ± 0.20 d	0.61 ± 0.03 c	1.34 ± 0.06 c
	M–SS	19.06 ± 0.38 c	15.82 ± 0.20 c	0.77 ± 0.08 c	1.72 ± 0.16 bc
XQ4-1	CK	25.83 ± 0.33 b	19.99 ± 0.33 b	1.30 ± 0.07 a	1.80 ± 0.10 b
	M–CK	28.27 ± 0.36 a	22.66 ± 0.67 a	1.45 ± 0.03 a	2.15 ± 0.06 a
	SS	19.96 ± 0.29 d	13.25 ± 0.49 d	0.69 ± 0.04 c	1.26 ± 0.05 c
	M–SS	23.50 ± 0.51 c	17.28 ± 0.59 c	1.02 ± 0.06 b	1.62 ± 0.12 b

Different lowercase letters indicated significant differences in *p* < 0.05 level. SL: seedling length; SLA: seedling leaf area; SFW: seedling fresh weight; RFW: root fresh weight. CK: 0 mM NaCl + 0 μM melatonin (MT) treatment; SS: 180 mM NaCl + 0 μM MT treatment; M–CK: 0 mM NaCl + 150 μM MT treatment; M–SS: 180 mM NaCl + 150 μM MT treatment.

## Data Availability

Data are contained within the article and Appendix A.

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
