# Peer review of "Exogenous Melatonin Alleviates NaCl Injury by Influencing Stomatal Morphology, Photosynthetic Performance, and Antioxidant Balance in Maize"

_ijms, 2024, doi:10.3390/ijms251810077_

Round 1
Reviewer 1 Report
Comments and Suggestions for Authors
The manuscript entitled “Exogenous Melatonin Alleviates NaCl Injury by Influencing Stomatal Morphology, Photosynthetic Performance and Antioxidant Balance in Maize” highlights the importance of melatonin (MT) in maize growth and its response to NaCl stress. However, there are significant concerns about ethics, language, and the results that are not discussed.
Line 18: “… stomatal morphologys”, delete “S”
Line 20: What do you mean by “corresponding gene”?
Lines 17 to 22: rewrite this long sentence into two clear sentences.
Line 23: “… induction” is inappropriate in this context. You induced salt stress and applied MT to alleviate the effects of stress.
Lines 22 to 30: What do you mean? Please write short, understandable, and concise sentences.
Line 31: What type of “gene” expression have you analyzed? Be specific
Line 34: “As a result”? What is the correlation between your previous sentence (gene expression) and this current one (plant growth and biomass)?
Line 42: “catastrophic”? The effects of salt stress can be severe and be qualified by catastrophic. But how catastrophic is the salinity itself?
Line 44: Review the meaning of the expression “Not altogether” in the context you have used.
Line 54: [… most crops “including” not especially Maize…]
Delete line 55. It does not make any sense unless you rewrite your sentence.
Lines 56 to 58: “China, the second largest maize producer worldwide, contributing 23.5% of global maize production and 42.2% of total cereal production”. This is an incomplete sentence.
Line 58 to 60: [Maize holds … soil salinization]. I understand your focus is the salt stress, but you can say, “Maize holds … challenged by climate changes and abiotic stress factors such as soil salinization.”
Line 60 and 61: [many chemicals … this purpose]. What purpose? What do you want to say?
Line 64: Delete “According to” and start your phrase with “Previous.”
Line 66 to 68: [As a pleiotropic … “the significance of MT”…]. What do you mean by “the significance of MT”? Do you mean MT function?
Line 70: Why did you capitalize each word in “Barley Plants’ Photosynthetic Efficiency”?
Lines 71 to 73: Rewrite this sentence
Line 74: [… “is” a promising …]. Replace it with “could be.”
Line 85 to 87: [These findings … ] You do not need this part in the introduction section.
Line 90: What is “SS”?
Lines 92, 93, and 95: What are the meanings of “CK,” “M-CK”, and “M-SS”? Please make sure to define all the abbreviations the first time that you mentioned them.
Line 98: “180 mM”. You can talk about the NaCl stress without mentioning the concentration. Concentration is important in the abstract, materials, and methods sections. Go through the manuscript and correct it.
Line 102: Delete “root”
Lines 111 and 112: [Several studies … in plants]. Delete this part. You are reporting your results, so talking about previous studies, you will need to cite the sources. This fit in the discussion section. Here, instead, you can tell the readers why it is important to run these analyses and then give the results.
Lines 139 to 143: Please split this into two concise sentences.
Lines 156 to 161: [Eight maize genotypes SOD activity… ], explain please.
Line 165: Please avoid using “to summarize” or “in summary” in the middle of results reports.
Line 178: [Effects of MT on Stomatal Morphology of Maize Seedlings under Salt Stress]. Please provide the microscopic aspect of the stomatal morphology in addition to Figure 4.
Lines 200 to 202: [… that salt stress inhibits … under salt stress]. Confusion sentence. Delete “under salt stress”
Line 239: “Photosynthetic Parameters,” do not capitalize the words. Same as in lines 243 and 252. Are there any reasons you capitalized these two words?
Lines 277 and 278: You said the candidate genes are associated with photosynthetic performance and antioxidant homeostasis and play a role under salt stress”. I am surprised that you did not discuss any of these genes. Or even you did not talk about their specific roles in Maize under NaCl stress conditions.
Lines 300 and 301: [eight maize genotypes SL … on average]. What are you trying to say?
Line 304: [This was in “agreement” with previous studies]. I don’t think “agreement” is the correct expression. Besides, you mentioned previous studies but ended up citing only one study.
Lines 307 to 311: [These findings … respectively]. Rewrite these two sentences.
Line 317: Why “In conclusion” in the middle of the discussion?
Line 318: [… photosynthetic enzymes …]. You are mixing up a lot. I have not seen any photosynthetic enzymes activity that you evaluated. If you are talking about SOD, POD CAT, and APX, they are all antioxidant enzymes present in plants.
Lines 324 and 325: [In addition, salt …. oxidative stress]. It is a fact that salt stress induces osmotic stress and ion toxicity, producing ROS as signaling molecules. You should cite previous works (Munns, Rana, and Mark Tester. “Mechanisms of salinity tolerance.” Annu. Rev. Plant Biol. 59.1 (2008): 651-681.; Yang, Yongqing, and Yan Guo. “Unraveling salt stress signaling in plants.” Journal of integrative plant biology 60.9 (2018): 796-804.; and many more).
Lines 331 to 333: [While … metabolic abnormalities]. This should be a sentence, not two.
Lines 343 to 352: Please move up this part to line 326 and delete [Increasing evidences … different abiotic stress].
Figures: In figures (1 to 7), please make sure that the letters on standard error bars are not overlapped.
Figure 8: In lines 357 and 358, you said, “We constructed a mechanistic map of the role of MT in the improvement salt resistance in maize seedlings under NaCl stress.” Not only is the quality of the figure terrible, but you did not construct the map. You literally copied and modified the figure published by Li, Z., Su, X., Chen, Y. et al. Melatonin Improves Drought Resistance in Maize Seedlings by Enhancing the Antioxidant System and Regulating Abscisic Acid Metabolism to Maintain Stomatal Opening Under PEG-Induced Drought. J. Plant Biol. 64, 299–312 (2021). https://doi.org/10.1007/s12374-021-09297-3 without citing the study.
This is wrong and can damage your credibility as researchers. I suggest the following steps:
1. Simply delete this figure from your manuscript
2. Create your own figure. If the figure is still inspired by Li et al. 2021 work, it’s ethical to acknowledge them by adding “Adapted from Li et al. 2021”. Do not just download the figure and edit it.
Line 367: Add “separately” before “sterilized.”
Line 381: “SPAD” is not the abbreviation of leaf greenness. Please provide the full meaning of SPAD.
Line 403: 4.4. Determination of antioxidant enzyme activity and H2O2 activity. Please rewrite this part in proper English. Writing it in a “telegraphic style” cannot help readers understand it.
Line 420: [… “we” carefully remove(d)] “we” is missing, and put the verb in the past tense.
In the whole part, look for any vocabulary mistakes and correct them.
Line 428: 4.6. Determination of relative water content and plasma membrane permeability. Rewrite in correct English this part. Do not write in “telegraphic style”.
Line 468: “… stomata opening…”. It is better to show the microscopic aspects of the stomata under the different treatments.
Line 472: “seedling” not “seeding”
Comments on the Quality of English Language
I found it difficult to read and understand the manuscript. So, please, before resubmitting the revised version, find a language editing service to correct any misunderstandings, expressions, and grammar mistakes.
Author Response
Dear Editor and Reviewers
Thank you for your letter of – and for the referee’s comments concerning our manuscript, “Exogenous Melatonin Alleviates NaCl Injury by Influencing Stomatal Morphology, Photosynthetic Performance and Antioxidant Balance in Maize (Manuscript ID: ijms-3181098)”. Wehave carefully studied these comments and have made corresponding corrections to the manuscript, which we describe in detail below. We would like to re-submit the manuscript and that for possible publication on the Special Issue: “Plant Development and Hormonal Signaling” of International Journal of Molecular Sciences. Thank you very much for your time and consideration.
Editor:
Your manuscript has now been reviewed by experts in the field and can be found with the review reports at: https://susy.mdpi.com/user/manuscripts/resubmit/46632d136f7d24e4f3ed160f1c2aeadf In total, there are 3 review reports for your manuscript. To expedite the process, we are sending you the 2 reports first, and we will most likely send you the remaining reviews in the next few days. If reviewers #3 do not upload a review within the next two days, we will cancel the review request. Let's hope that's fine with you. Please revise the manuscript found at the above link according to the reviewers' comments and upload the revised file within 10 days. Note the following check-list:
Thanks for the positive comments of you and all reviewers for our manuscript. As suggested, we have carefully revised and improved our manuscript using the “Track Changes” function of the manuscript at the above link. We then have re-submitted the manuscript within the allotted time.
Thank you for your consideration.
(I) Ensure all references are relevant to the content of the manuscript.
Thanks for the positive comments. As suggested, we have carefully checked all references. We then have re-submitted the manuscript.
Thank you for your consideration.
(II) Highlight any revisions to the manuscript, so editors and reviewers can see any changes made.
Thanks for the positive comments. As suggested, we have carefully revised and improved our manuscript using the “Track Changes” function of the manuscript. We then have re-submitted the manuscript.
Thank you for your consideration.
(III) Provide a cover letter to respond to the reviewers’comments and explain, point by point, the details of the manuscript revisions.
Thanks for your positive comments for our manuscript. As suggested, we have carefully revised and improved our manuscript. In addition, we have prepared a detailed response letter to all reviewers for each point, and then have re-submitted the manuscript.
Thank you for your consideration.
(IV) If the reviewer(s) recommended references, critically analyze them to ensure that their inclusion would enhance your manuscript. If you believe these references are unnecessary, you should not include them.
Thanks for your positive comments for our manuscript. As suggested, we have carefully checked and revised the References. At the same time, we also have re-added new references to enhance the quality of our manuscript. We then have re-submitted the manuscript.
Thank you for your consideration.
(V) If you found it impossible to address certain comments in the review reports, include an explanation in your appeal.
Thanks for your positive comments for our manuscript. As suggested, we have carefully revised and improved our manuscript. In addition, we have prepared a detailed response letter to all reviewers for each point, and then have re-submitted the manuscript.
Thank you for your consideration.
We would like to draw your attention to the status of this invitation“Publish Author Biography on the webpage of the paper”https://susy.mdpi.com/user/manuscript/author_biography/46632 d136f7d24e4f3ed160f1c2aeadf. If you decide to publish your biography, please remember to fill in it before your paper is accepted. If your manuscript requires improvement to the language and/or figures, you may consider MDPI Author Services: https://www.mdpi.com/authors/english.
Thanks for the positive comments. As suggested, we have carefully checked and revised the English language of the manuscript. We then re-submitted the manuscript.
In addition, thanks for your invitation, we decided not to publish our biography.
Thank you for your consideration.
Please do not hesitate to contact us if you have any questions regarding the revision of your manuscript or if you need more time. We look forward to hearing from you soon.
Thanks for your positive comments for our manuscript. As suggested, we have carefully revised and improved the manuscript using the “Track Changes” function of our manuscript at the above link. We then have re-submitted the manuscript within the allotted time.
Thank you for your consideration.
Reviewer 1:
The manuscript entitled “Exogenous Melatonin Alleviates NaCl Injury by Influencing Stomatal Morphology, Photosynthetic Performance and Antioxidant Balance in Maize” highlights the importance of melatonin (MT) in maize growth and its response to NaCl stress. However, there are significant concerns about ethics, language, and the results that are not discussed.
Thanks for your positive comments. As suggested, we have made serious changes in ethics, language and results. We then re-submitted the manuscript.
Thank you for your consideration.
- Line 18: “… stomatal morphologys”, delete “S”
Thanks for your positive comments. As suggested, we have revised the corresponding contents were that “stomatal morphology” in Line 19 of the manuscript. We then have re-submitted the manuscript.
Thank you for your consideration.
- Line 20: What do you mean by “corresponding gene”?
Thanks for your positive comments. As suggested, “the corresponding genes” were superoxide dismutase [Mn] 3.4, mitochondria (Zm00001d009990), superoxide dismutase [Cu-Zn] 4AP (Zm00001d047479), catalase 1 (Zm00001d014848), ascorbate peroxidase 2 (Zm00001d007234), chlorophyll a oxygenase chloroplast (Zm00001d011819), and 9 cisepoxide carotenoid dioxygenase 8 (Zm00001d017766) (Figure 7; Table S1). Therefore, the corresponding contents were revised that “meanwhile the six gene expression levels controlling antioxidant enzyme activities and photosynthetic pigments biosynthesis in two materials with contrasting salt resistance were examined at all treatments, to explore the possible molecular mechanism of exogenous MT alleviating salt injury in maize.” in Lines 20-23 of the manuscript. We then have re-submitted the manuscript.
Thank you for your consideration.
- Lines 17 to 22: rewrite this long sentence into two clear sentences.
Thanks for your positive comments. As suggested, we have revised the corresponding contents were that “Herein, we investigated the effects of 150 μM exogenous MT on multiple phenotypes and physiologic metabolisms in three-leaf seedlings across eight maize inbred lines under 180 mM NaCl salt stress, including growth parameters, stomatal morphology, photosynthetic metabolisms, antioxidant enzyme activities, and reactive oxygen species (ROS); meanwhile the six gene expression levels controlling antioxidant enzyme activities and photosynthetic pigments biosynthesis in two materials with contrasting salt resistance were examined at all treatments, to explore the possible molecular mechanism of exogenous MT alleviating salt injury in maize.” in Lines 17-25 of the manuscript. We then have re-submitted the manuscript.
Thank you for your consideration.
- Line 23: “… induction” is inappropriate in this context. You induced salt stress and applied MT to alleviate the effects of stress.
Thanks for your positive comments. As suggested, the “induction” has been replaced to the “application” in Line 25 of the manuscript. We then have re-submitted the manuscript.
Thank you for your consideration.
- Lines 22 to 30: What do you mean? Please write short, understandable, and concise sentences.
Thanks for your positive comments. As suggested, we have further revised the corresponding contents, namely: “The results showed that 150 μM exogenous MT application protected membrane integrity and reduced ROS accumulation by activating antioxidant system in leaves of maize seedlings under salt stress, their relative conductivity and H2O2 level average reduced by 20.91% and 17.22%, while the activities of superoxide dismutase (SOD), peroxidase (POD), catalase (CAT), and ascorbate peroxidase (APX) averaged increased by 13.90%, 17.02%, 22.00%, and 14.24% relative to compared with salt stress alone, respectively. The improvement of stomatal size and the deposition of photosynthetic pigments were more favorable to enhancing photosynthesis in leaves when these seedlings treated with MT application under salt stress, their stomotal size, chlorophyll a content, and net photosynthetic rate averaged increased by 11.60%, 19.64%, and 27.62%, respectively.” in Lines 25-34 of the manuscript. We then have re-submitted the manuscript.
Thank you for your consideration.
- Line 31: What type of “gene” expression have you analyzed? Be specific
Thanks for your positive comments. As suggested, we have described the type of genes, namely: “the expression of antioxidant enzyme genes (Zm00001d009990, Zm00001d047479, Zm00001d014848, and Zm00001d007234) and photosynthetic pigment biosynthesis genes (Zm00001d011819 and Zm00001d017766)” in Lines 35-37 of the manuscript. We then have re-submitted the manuscript.
Thank you for your consideration.
- Line 34: “As a result”? What is the correlation between your previous sentence (gene expression) and this current one (plant growth and biomass)?
Thanks for your positive comments. As suggested, we have revised the corresponding contents, namely: “At the same time, 150 μM MT significantly promoted seedling growth and biomass accumulation.” in Lines 37-38 of the manuscript. We then have re-submitted the manuscript.
Thank you for your consideration.
- Line 42: “catastrophic”? The effects of salt stress can be severe and be qualified by catastrophic. But how catastrophic is the salinity itself?
Thanks for your positive comments. As suggested, we have revised the corresponding contents, namely: “Soil salinity is one of serious issue among all abiotic stresses as it severely limits agricultural production across the globe [1].” in Lines 45-46 of the manuscript. We then have re-submitted the manuscript.
Thank you for your consideration.
- Line 44: Review the meaning of the expression “Not altogether” in the context you have used.
Thanks for your positive comments. As suggested, we have corrected the contents, namely: “Salinity has devastatingly affected the area of more than 1.2 billion hectares [2], and high salinity is estimated to affect 20% and 33% of total cultivated and irrigated agricultural lands worldwide.” in Lines 46-49 of the manuscript. We then have re-submitted the manuscript.
Thank you for your consideration.
- Line 54: [… most crops “including” not especially Maize…]
Thanks for your positive comments. As suggested, we have corrected the contents, namely: “Notably, most crops are salt-sensitive.” in Lines 57-58 of the manuscript. We then have re-submitted the manuscript.
Thank you for your consideration.
- Delete line 55. It does not make any sense unless you rewrite your sentence.
Thanks for your positive comments. As suggested, we have deleted the corresponding contents. namely: “Notably, most crops, especially maize (Zea mays L.), are salt-sensitive, prompting extensive studies to uncover the mechanism of salt tolerance in maize.” corrected “Notably, most crops are salt-sensitive.” in Lines 57-58 of the manuscript. We then have re-submitted the manuscript.
Thank you for your consideration.
- Lines 56 to 58: “China, the second largest maize producer worldwide, contributing 23.5% of global maize production and 42.2% of total cereal production”. This is an incomplete sentence.
Thanks for your positive comments. As suggested, we have corrected the contents, namely: “China, the second largest maize producer worldwide, contributing 23.5% of global maize production and 42.2% of total cereal production, which plays an important role in ensuring food security [9].” in Lines 60-63 of the manuscript. We then have re-submitted the manuscript.
Thank you for your consideration.
- Line 58 to 60: [Maize holds … soil salinization]. I understand your focus is the salt stress, but you can say, “Maize holds … challenged by climate changes and abiotic stress factors such as soil salinization.”
Thanks for your positive comments. As suggested, we have corrected the contents, namely: “Maize holds challenged by climate changes and abiotic stress factors such as soil salinization [10].” in Lines 64-65 of the manuscript. We then have re-submitted the manuscript.
Thank you for your consideration.
- Line 60 and 61: [many chemicals … this purpose]. What purpose? What do you want to say?
Thanks for your positive comments. As suggested, many chemicals, reagents, hormones, and growth regulators have been developed to deal with the harm of soil salinization to maize. we have corrected the contents, namely: “Fortunaltly, many chemicals, reagents, hormones, and growth regulators have been developed to deal with the harm of soil salinization to maize.” in Lines 65-67 of the manuscript. We then have re-submitted the manuscript.
Thank you for your consideration.
- Line 64: Delete “According to” and start your phrase with “Previous.”
Thanks for your positive comments. As suggested, we have corrected the contents, namely: “Previous studies proved that MT could improve plant resistance against various environmental stresses, including salt, heavy metal, drought, and high/low temperature [14–16].” in Lines 71-73 of the manuscript. We then have re-submitted the manuscript.
Thank you for your consideration.
- Line 66 to 68: [As a pleiotropic … “the significance of MT”…]. What do you mean by “the significance of MT”? Do you mean MT function?
Thanks for your positive comments. Yes, where mean the MT function. Because the sentence A “As a pleiotropic nontoxic signaling molecule, universal bio-stimulator and essential metabolite, the significance of MT is gradually getting established in plants [17] (Lines 73-75)” and sentence B “Melatonin (MT), an indole compound, first identified in 1995, and found in both plants and animals, has been revealed to play important roles in growth and development, and stress response of plants [11–13]. (Lines 69-71)” were repeated, we have deleted sentence A. We then have re-submitted the manuscript.
Thank you for your consideration.
- Line 70: Why did you capitalize each word in “Barley Plants’ Photosynthetic Efficiency”?
Thanks for your positive comments. As suggested, we have corrected the contents, namely: “Under salt stress, MT treatments increase barley plants’photosynthetic efficiency.” in Lines 76-77 of the manuscript. We then have re-submitted the manuscript.
Thank you for your consideration.
- Lines 71 to 73: Rewrite this sentence
Thanks for your positive comments. As suggested, we have corrected the contents, namely: “Under saline conditions, significant increases appeared in leaf area, biomass, and photosynthesis efficiency of Brassica juncea L. with MT treatments [20].” in Lines 79-81 of the manuscript. We then have re-submitted the manuscript.
Thank you for your consideration.
- Line 74: [… “is” a promising …]. Replace it with “could be.”
Thanks for your positive comments. As suggested, we have revised and improved the corresponding content, namely: “Thus, exogenous MT application could be approach to alleviate stress injury in multiple plant species [21].” in Lines 81-82 of the manuscript. We then have re-submitted the manuscript.
Thank you for your consideration.
- Line 85 to 87: [These findings … ] You do not need this part in the introduction section.
Thanks for your positive comments. As suggested, we have revised the content, namely: “Which will provide a valuable reference for improving salt resistance in maize, thus maintaining yield output in maize production in coping with the global salinization process.” in Lines 92-95 of the manuscript. We then have re-submitted the manuscript.
Thank you for your consideration.
- Line 90: What is “SS”?
Thanks for your positive comments. As suggested, “SS” is “180 mM NaCl + 0 μM MT treatment”. Therefore, we have revised and improved the content, namely: “Under 180 mM NaCl+0 μM MT treatment (SS)” in Line 98 of the manuscript. We then have re-submitted the manuscript.
Thank you for your consideration.
- Lines 92, 93, and 95: What are the meanings of “CK,” “M-CK”, and “M-SS”? Please make sure to define all the abbreviations the first time that you mentioned them.
Thanks for your positive comments. As suggested, “CK” is “0 mM NaCl+0 μM MT treatment”. “SS” is “180 mM NaCl+0 μM MT treatment”. “M–CK” is “0 mM NaCl+150 μM MT treatment”. “M–SS” is “180 mM NaCl+150 μM MT treatment”. Therefore, we have revised and improved the content, namely: “Under 180 mM NaCl+0 μM MT treatment (SS), the seedling length (SL), seedling leaf area (SLA), seeding fresh weight (SFW), and root fresh weight (RFW) of eight maize varieties were less than that was treated with 0 mM NaCl+0 μM MT treatment (CK) (Table 1). Significantly reduced by an average of 23.26%, 33.24%, 37.32, and 28.89%, respectively. The findings indicated that NaCl stress had a negative effect on seedling growth and biomass accumulation in maize. These four traits were increased in 0 mM NaCl+150 μM MT treatment (M–CK) compared to CK by 10.08%, 16.60%, 16.73%, and 18.10%, respectively (Table 1). At the same time, they were increased in 180 mM NaCl+150 μM MT treatment (M-SS) compared to SS treatment, and significantly increased by 17.01%, 28.61%, 32.32%, and 25.01%, respectively (Table 1).” in Lines 98-107 of the manuscript. We then have re-submitted the manuscript.
Thank you for your consideration.
- Line 98: “180 mM”. You can talk about the NaCl stress without mentioning the concentration. Concentration is important in the abstract, materials, and methods sections. Go through the manuscript and correct it.
Thanks for your positive comments. As suggested, we agree with you very much and have deleted "180 mM ", in addition, we have also made corresponding modifications. We have also made corresponding modifications We then have re-submitted the manuscript.
Thank you for your consideration.
- Line 102: Delete “root”
Thanks for your positive comments. As suggested, we have deleted the corresponding contents, namely:“Under 180 mM NaCl+0 μM MT treatment (SS), the seedling length (SL), seedling leaf area (SLA), seeding fresh weight (SFW), and root fresh weight (RFW) of eight maize varieties were less than that was treated with 0 mM NaCl+0 μM MT treatment (CK) (Table 1). Significantly reduced by an average of 23.26%, 33.24%, 37.32, and 28.89%, respectively. Indicating that NaCl stress had a negative effect on seedling growth and biomass accumulation in maize. These four traits were increased in 0 mM NaCl+150 μM MT treatment (M–CK) compared to CK by 10.08%, 16.60%, 16.73%, and 18.10%, respectively (Table 1). At the same time, they were increased in 180 mM NaCl+150 μM MT treatment (M-SS) compared to SS treatment, and significantly increased by 17.01%, 28.61%, 32.32%, and 25.01%, respectively (Table 1). Suggesting that MT application could significantly alleviate salt injury of maize seedlings to maintain overall health and growth potential.”in Lines 101-113 of the manuscript. We then have re-submitted the manuscript.
Thank you for your consideration.
- Lines 111 and 112: [Several studies … in plants]. Delete this part. You are reporting your results, so talking about previous studies, you will need to cite the sources. This fit in the discussion section. Here, instead, you can tell the readers why it is important to run these analyses and then give the results.
Thanks for your positive comments. As suggested, we have revised the content, namely: “Photosynthetic parameters can reflect the physiological status and changes of maize seedlings under salt stress [38].” in Lines 128-129 of the manuscript. We then have re-submitted the manuscript.
Thank you for your consideration.
- Lines 139 to 143: Please split this into two concise sentences.
Thanks for your positive comments. As suggested, we have revised the content, namely: “The result shows that 180 mM NaCl stress inhibits maize Chl a, Chl b, Car production and decreases SPAD. The application of 150 μM exogenous MT can alleviate the breakage of maize chlorophyll by salt stress, maintain higher photosynthesis during maize seedling development, enhance light absorption and transmission, and effectively reduce the damage to maize.” in Lines 170-174 of the manuscript. We have modified the content. We then have re-submitted the manuscript.
Thank you for your consideration.
- Lines 156 to 161: [Eight maize genotypes SOD activity… ], explain please.
Thanks for your positive comments. As suggested, we have revised the content, namely:“Moreover, M–CK treated leaves of the eight maize varieties, which showed a 20.15% increase with the average SOD activity, a 22.96% increase with average POD activity, a 36.26% increase with average CAT activity, and a 13.87% increase with average APX activity compared to CK treatment (Figure 3).” in Lines 204-207 of the manuscript. We then have re-submitted the manuscript.
Thank you for your consideration.
- Line 165: Please avoid using “to summarize” or “in summary” in the middle of results reports.
Thanks for your positive comments. As suggested, we have deleted the corresponding results reports contents and revised the content, namely: For the leaves of all maize seedlings under SS treatment, the superoxide dismutase (SOD) activity significantly increased by 16.96% to 82.03% compared to CK, except for GS36–2, the peroxidase (POD) activity significantly increased by 18.30% to 106.00%, the catalase (CAT) activity significantly increased by 48.97% to 163%, and the ascorbate peroxidase (APX) activity significantly increased by 17.86% to 42.75% (Figure 3). Moreover, M–CK treated leaves of the eight maize varieties, which showed a 20.15% increase with the average SOD activity, a 22.96% increase with average POD activity, a 36.26% increase with average CAT activity, and a 13.87% increase with average APX activity compared to CK treatment (Figure 3). Like M–CK treatment, the all seedlings were cultured in M-SS treatment, the activities of four antioxidant enzymes had significant increase with 13.90%, 17.02%, 22.01%, and 14.24%, respectively (Figure 3). In addition, the H2O2 content of all maize seedlings displayed significant increase (ranging from 21.26% to 62.62%) under SS treatment that were relative to CK treated seedlings; while which had significant decrease under M–CK (27.96~52.77%) and M–SS (8.85~24.49%) treatments compared to SS treatment, respectively (Figure 3).” in Lines 199-224 of the manuscript. We then have re-submitted the manuscript.
Thank you for your consideration.
- Line 178: [Effects of MT on Stomatal Morphology of Maize Seedlings under Salt Stress]. Please provide the microscopic aspect of the stomatal morphology in addition to Figure 4.
Thanks for your positive comments. As suggested, we provide a map of the microscopic aspects of stomatal morphology S1 as follows.
Thank you for your consideration.
Figure S1. Stomatal morphological characteristics of adaxial stomata of (A, C) and abaxial stomata (B, D) of the 3rd leaf in maize seedlings under different treatments. A and C were the adaxial stomata of LX19 and NX40-6 leaves, respectively. B and D were the abaxial stomata of LX19 and NX40–6 leaves. Thea, b, c and d were the four treatments including CK, M–CK, SS, and M–SS respectively. CK: 0 mM NaCl+0 μM melatonin (MT) treatment; SS: 180 mM NaCl+0 μM MT treatment; M–CK: 0 mM NaCl+150 μM MT treatment; M–SS: 180 mM NaCl+150 μM MT treatment.
- Lines 200 to 202: [… that salt stress inhibits … under salt stress]. Confusion sentence. Delete “under salt stress”
Thanks for your positive comments. As suggested, we have deleted the corresponding results reports contents. and revised the content, namely: “There were no significant differences for stomatal length of adaxial surface (SL–Z) and stomatal length of abaxial surface (SL–F) of leaves in eight maize varieties under both SS and M–CK treatment compared to CK (Figure 4A; 4B). SS treatment significantly reduced the stomatal width of adaxial surface (SW–Z) and stomatal width of abaxial surface (SW–F) of these maize leaves compared to CK (Figure 4C; 4D), their SW–Z clearly reduced by 7.50~18.71% (Figure 4C) and their SW-F clearly reduced by 10.52~24.37% (Figure 4D). The SW–Z and SW–F of all maize in M–CK treatment were significantly higher than that of CK, with the significant increase of 9.41~21.26% in SW–Z (Figure 4C) and 7.50% to 16.83% in SW–F (Figure 4D). M–SS treatment significantly increased the SW–Z and SW–F compared to SS treatment, with a significant increase in SW–Z ranging from 7.89% to 22.78% (Figure 4C). SW–F of XY201 was slightly increased, the average SW–F of seven seedlings significantly increased by 10.45%. Compared with CK, the stomatal area of adaxial surface (SA–Z) and stomatal area of abaxial surface (SA–F) of SS treated maize seedlings also significantly reduced (Figure 4E; 4F), with the SA-Z significantly reduced by 12.03~25.64% (Figure 4E), and the SA-F significantly reduced by 11.76~26.12% (Figure 4F). The SA–Z and SA–F of all maize seedlings in M–CK treatment were higher than that of CK, which showed 14.64% and 18.86% increase, respectively (Figure 4E; 4F). Additionally, compared with SS treatment, the SA–Z had 10.02~21.10% increase and the SA–F had 6.90~13.00% (except for NX40–6) increase under M–SS treatment (Figure 4E; 4F).” in Lines 237-269 of the manuscript. We then have re-submitted the manuscript.
Thank you for your consideration.
- Line 239: “Photosynthetic Parameters,” do not capitalize the words. Same as in lines 243 and 252. Are there any reasons you capitalized these two words?
Thanks for your positive comments. As suggested, we have revised the content, namely: “To further understand the complex relationship networks of 26 traits and six candidate genes expression, their Pearson correlation analysis was performed across LX19 and NX40-6 leaves under four treatments (Figure 7). Interestingly, there were 207 groups with significant (p < 0.01 or p < 0.05) correlations between both traits, 15 groups with significant (p < 0.01 or p < 0.05) correlations between both genes, as well as 119 groups with significant (p < 0.01 or p < 0.05) correlations between trait and gene (Figure 7). Such as, the expression levels of antioxidant enzyme gene (Zm00001d047479) showed significantly positive correlation to REC, Ci, SOD, POD, CAT, APX, and H2O2 and showed significantly negative correlation to SL, SLA, SFW, RFW, RWC, Pn, Gs, Tr, SL–F, SW–F, SA–F, SW–Z, Chl a, Chl b, Chl a/b, and SPAD. The expression levels of photosynthetic pigments biosynthetic gene (Zm00001d017766) showed significantly positive correlation to SL, RFW, Pn, Gs, Tr, SW–F, SA–F, SW–Z, Chl a, Chl b, Car, Chla/b, and SPAD and showed significantly negative correlation to REC, Ci, SOD, CAT, APX, and H2O2 (Figure 7). These findings thus showed that the six candidate genes for antioxidant enzyme activity and photosynthetic pigment biosynthesis could cooperate with each other to directly or indirectly regulate multiple metabolism and development processes including seedling growth, stomatal morphology, photosynthetic performance, antioxidant system, ROS homeostasis, and membrane integrity in maize under diverse environment.” in Lines 343-364 of the manuscript. We then have re-submitted the manuscript.
Thank you for your consideration.
- Lines 277 and 278: You said the candidate genes are associated with photosynthetic performance and antioxidant homeostasis and play a role under salt stress”. I am surprised that you did not discuss any of these genes. Or even you did not talk about their specific roles in Maize under NaCl stress conditions.
Thanks for your positive comments. As suggested, we have revised the content, namely: “The results showed that the expression levels of four antioxidant enzyme genes significantly enhanced under SS stress of LX19 and NX40-6 leaves compared to CK treatment with 0.44-~64-fold and 0.77~77-fold (Figure 6), while the expression of two photosynthetic pigment biosynthesis genes significantly reduced with 23.59~31.25% and 30.23~31.48% (Figure 6); Further the M-SS treatment induced upregulation of all genes expression levels, with 2.93~20.81% and 5.47~35.78% compared to SS stress (Figure 6).” in Lines 324-330 of the manuscript. We then have re-submitted the manuscript.
Thank you for your consideration.
- Lines 300 and 301: [eight maize genotypes SL … on average]. What are you trying to say?
Thanks for your positive comments. As suggested, we have revised the content, namely: “In this study, we found that the growth of maize was notably inhibited when exposed to salt stress, the average SL, SLA, SFW, and RFW of eight maize varieties showed significant reduced by 23.26%, 33.24%, 37.32%, and 28.89%, respectively.” in Lines 426-429 of the manuscript. We then have re-submitted the manuscript.
Thank you for your consideration.
- Line 304: [This was in “agreement” with previous studies]. I don’t think “agreement” is the correct expression. Besides, you mentioned previous studies but ended up citing only one study.
Thanks for your positive comments. As suggested, we have revised the content, namely: “This was in consistent with previous studies [27,28,31].” in Lines 432-433 of the manuscript. We then have re-submitted the manuscript.
Thank you for your consideration.
- Lines 307 to 311: [These findings … respectively]. Rewrite these two sentences.
Thanks for your positive comments. As suggested, we have revised the content, namely: “Our results showed that 150 μM exogenous MT application caused significant increase in contents of Chl a (19.64%), Chl b (13.44%), and Car (27.61%) in maize seedlings under 180 mM salt stress, respectively (Figure 2A; 2B; 2D)” in Lines 436-440 of the manuscript. We then have re-submitted the manuscript.
Thank you for your consideration.
- Line 317: Why “In conclusion” in the middle of the discussion?
Thanks for your positive comments. As suggested, we have revised the content, namely: “Therefore, exogenous MT may induce photosynthetic pigments biosynthesis, improve the photosystem, activate antioxidant enzyme activities, as well as regulate stomatal morphology, to maintain the normal photosynthetic process, resulting in Pn increased by 27.62% and Gs increased by 35.73% in leaves of all maize seedling that treated with 150 μM exogenous MT application under 180 mM NaCl stress.” in Lines 451-456 of the manuscript. We then have re-submitted the manuscript.
Thank you for your consideration.
- Line 318: [… photosynthetic enzymes …]. You are mixing up a lot. I have not seen any photosynthetic enzymes activity that you evaluated. If you are talking about SOD, POD CAT, and APX, they are all antioxidant enzymes present in plants.
Thanks for your positive comments. As suggested, we have revised the content, namely: “Therefore, exogenous MT may induce photosynthetic pigments biosynthesis, improve the photosystem, activate antioxidant enzyme activities, as well as regulate stomatal morphology, to maintain the normal photosynthetic process, resulting in Pn increased by 27.62% and Gs increased by 35.73% in leaves of all maize seedling that treated with 150 μM exogenous MT application under 180 mM NaCl stress.” in Lines 451-456 of the manuscript. We then have re-submitted the manuscript.
Thank you for your consideration.
- Lines 324 and 325: [In addition, salt …. oxidative stress]. It is a fact that salt stress induces osmotic stress and ion toxicity, producing ROS as signaling molecules. You should cite previous works (Munns, Rana, and Mark Tester. “Mechanisms of salinity tolerance.” Annu. Rev. Plant Biol. 59.1 (2008): 651-681.; Yang, Yongqing, and Yan Guo. “Unraveling salt stress signaling in plants.” Journal of integrative plant biology 60.9 (2018): 796-804.; and many more).
Thanks for your positive comments. As suggested, we cite and add relevant literature: “
Munns, R.; Tester, M. Mechanisms of salinity tolerance. Annu. Rev. Plant Biol. 2008, 59, 651-681.
Yang, Y.Q.; Guo, Y. Unraveling salt stress signaling in plants. J. Integr. Plant Biol. 2018, 60, 796-804.
Arif, Y.; Singh, P.; Siddiqui, H.; Bajguz, A.; Hayat, S. Salinity induced physiological and biochemical changes in plants: An omic approach towards salt stress tolerance. Plant Physiol. Biochem. 2020, 156, 64-77.” in Lines 743-746 of the manuscript. We then have re-submitted the manuscript.
Thank you for your consideration.
- Lines 331 to 333: [While … metabolic abnormalities]. This should be a sentence, not two.
Thanks for your positive comments. As suggested, we have revised the content, namely: “Appropriate concentrations of ROS are necessary signaling molecules in plants, on a hand, excessive ROS can cause lipid peroxidation, membrane disruption, on the other hand, which can lead to enzyme inactivation, and metabolic abnormalities [33–35].” in Lines 476-479 of the manuscript. We then have re-submitted the manuscript.
Thank you for your consideration.
- Lines 343 to 352: Please move up this part to line 326 and delete [Increasing evidences … different abiotic stress].
Thanks for your positive comments. As suggested, we have moved and revised the content. namely: “To reduce stress–triggered ROS accumulation, plants have evolved an effective antioxidant system, including enzymatic and non-enzymatic antioxidants. In plants, MT has also been suggested to be a crucial antioxidant that can scavenge oxygen free radicals effectively [26]. Many studies have indicated that exogenous MT application can increase some antioxidant enzyme (such as POD, SOD, and APX) activities under abiotic stress in plants [18,31,36]. In this experiment, we also found that the activities of SOD, POD, CAT, and APX were higher in the presence of MT under salt stress, and the expression of antioxidant enzyme genes (Zm00001d009990, Zm00001d047479, Zm00001d014848, and Zm00001d007234) were significantly enhanced, which may be mainly due to the synergistic effect of salt stress and exogenous MT. Therefore, the positive effects of exogenous MT on the active oxygen scavenging system could improve stress resistance of maize seedlings.” in Lines 469-480 of the manuscript. We then have re-submitted the manuscript.
Thank you for your consideration.
- Figures: In figures (1 to 7), please make sure that the letters on standard error bars are not overlapped.
Thanks for your positive comments. As suggested, we have modified the content and ensure that the letters on standard error bars are not overlapped. We then have re-submitted the manuscript.
Thank you for your consideration.
- Figure 8: In lines 357 and 358, you said, “We constructed a mechanistic map of the role of MT in the improvement salt resistance in maize seedlings under NaCl stress.” Not only is the quality of the figure terrible, but you did not construct the map. You literally copied and modified the figure published by Li, Z., Su, X., Chen, Y. et al. Melatonin Improves Drought Resistance in Maize Seedlings by Enhancing the Antioxidant System and Regulating Abscisic Acid Metabolism to Maintain Stomatal Opening Under PEG-Induced Drought. J. Plant Biol. 64, 299–312 (2021). https://doi.org/10.1007/s12374-021-09297-3 without citing the study.
This is wrong and can damage your credibility as researchers. I suggest the following steps:
- Simply delete this figure from your manuscript
- Create your own figure. If the figure is still inspired by Li et al. 2021 work, it’s ethical to acknowledge them by adding “Adapted from Li et al. 2021”. Do not just download the figure and edit it.
Thanks for your positive comments. As suggested, we deleted this figure and constructed a new figure. We then have re-submitted the manuscript.
Thank you for your consideration.
Figure 8. Schematic model of response mechanism for exogenous melatonin (MT) alleviates salt injury in maize seedlings. Solid arrows indicated promote effects, dashed arrows indicated inhibited effects; red arrows indicated positive expression of genes; ROS: reactive oxygen species; SOD: superoxide dismutase; POD: peroxidase, CAT: catalase; APX: ascorbate peroxidase; Pn; net photosynthetic rate; Gs: stomatal conductance; Tr: transpiration rate.
- Line 367: Add “separately” before “sterilized.”
Thanks for your positive comments. As suggested, we have revised the content, namely: “The random eight elite maize genotypes, including XY201, LX19, GS36–2, ZC2–7, NX32–1, NX40–6, GZ13–2, and XQ4–1 from Molecular Biology Laboratory, Stata Key Laboratory of Aridland Crop Science, China, were used in this study. The seeds of eight genotypes were separately sterilized with 0.5% (v/v) sodium hypochlorite solution for 15 min, rinsed five times with double–distilled water (ddH2O), and soaked in ddH2O for 24 h at 22±0.5°C indoor environment.” in Line 527 of the manuscript. We then have re-submitted the manuscript.
Thank you for your consideration.
- Line 381: “SPAD” is not the abbreviation of leaf greenness. Please provide the full meaning of SPAD.
Thanks for your positive comments. As suggested, we have revised the content, namely: “The Soil and relative chlorophyll content (SPAD) (Liu, S.C.; Xiong, Z; Zhang, Z.L.; Wei, Y.B.; Xiong, D.L. Exploration of chlorophyll fluorescence characteristics gene regulatory in rice (Oryza sativa L.): a genome-wide association study. Frontiers in Plant Science 2023, 1234866-1234866.) of the 3rd leaf of corresponding maize seedlings under all treatments was determined by the chlorophyll meter (SPAD-502, Japan). in Line 542 of the manuscript. We then have re-submitted the manuscript.
Thank you for your consideration.
- Line 403: 4.4. Determination of antioxidant enzyme activity and H2O2 activity. Please rewrite this part in proper English. Writing it in a “telegraphic style” cannot help readers understand it.
Thanks for your positive comments. As suggested, we have revised the content, namely: “Refer to the method of Zhao et al. [40], the SOD activity, POD activity, CAT activity, APX activity, and H2O2 content were determined using corresponding Solarbio kits (Beijing Solarbio Science and Technology Co., Ltd., Beijing, China) and using the multi-function microplate reader (SynergyHTX; BioTek Instruments, Inc. USA), following the manufacturer’s kit instructions, respectively.” in Lines 574-578 of the manuscript. We have also made modifications throughout the paper. We then have re-submitted the manuscript.
Thank you for your consideration.
- Line 420: [… “we” carefully remove(d)] “we” is missing, and put the verb in the past tense.
In the whole part, look for any vocabulary mistakes and correct them.
Thanks for your positive comments. As suggested, we have revised the content, namely: “The 3rd fresh leaf of maize seedlings under each treatment was cut and the middle of leaf (1cm ×1 cm) was sticked the slide coated with glue. Peeling off the adaxial and abaxial surfaces after 20 min. Stomatal morphology was observed using a forward and inverted integrated fluorescence microscope (Revolve RVL–100–G,ECHO, USA). Five fields of view were randomly selected for each treatment on the front and back sides. Stomatal length, width, and area were measured and statistically recorded under a 4× objective (image size 3226×3024).” in Lines 595-605 of the manuscript. We then have re-submitted the manuscript.
Thank you for your consideration.
- Line 428: 4.6. Determination of relative water content and plasma membrane permeability. Rewrite in correct English this part. Do not write in “telegraphic style”.
Thanks for your positive comments. As suggested, we have revised the content, namely: “The 1.0 g fresh leaves (FW) of the seedlings under each treatment was measured, and which then were completely immersed in 30 mL ddH2O for 12 h until constant water (TW). The leaves were then dried in an oven until the constant weight (DW). The relative water content (RWC) was calculated as follows: [(FW–DW)/(TW–DW)]×100%. ” in Lines 608-617 of the manuscript.
“The 0.1 g fresh leaves of the seedlings under each treatment were placed into the test tubes containing 10 mL ddH2O and soaked at room temperature for 12 h. The conductivity (R1) was measured by DDSJ-308F conductivity meter (Rex Electric Chemical, Shanghai, China). Then, the same set of samples were stored in a 100°C water bath for 15 min and the electrical conductivity (R2) was recorded. The REC was estimated as follows [40]: REC = (R1 / R2)×100%. ” in Lines 618-622 of the manuscript. We then have re-submitted the manuscript.
Thank you for your consideration.
- Line 468: “… stomata opening…”. It is better to show the microscopic aspects of the stomata under the different treatments.
Thanks for your positive comments. As suggested, we have revised the content, namely: “MT activated photosynthetic pigment biosynthesis genes expression (Zm00001d011819 and Zm00001d017766), prevented chlorophyll degradation, and improved stomata morphology, thus improving photosynthetic capacity under NaCl stress.” in Lines 631-635 of the manuscript. We provide a map of the microscopic aspects of stomatal morphology S1 as follows. We then have re-submitted the manuscript.
Thank you for your consideration.
Figure S1. Stomatal morphological characteristics of adaxial stomata of (A, C) and abaxial stomata (B, D) of the 3rd leaf in maize seedlings under different treatments. A and C were the adaxial stomata of LX19 and NX40-6 leaves, respectively. B and D were the abaxial stomata of LX19 and NX40–6 leaves. Thea, b, c and d were the four treatments including CK, M–CK, SS, and M–SS respectively. CK: 0 mM NaCl+0 μM melatonin (MT) treatment; SS: 180 mM NaCl+0 μM MT treatment; M–CK: 0 mM NaCl+150 μM MT treatment; M–SS: 180 mM NaCl+150 μM MT treatment.
- Line 472: “seedling” not “seeding”
Thanks for your positive comments. As suggested, we have revised the content, namely: “Overall, the application of exogenous MT has the potential to improve maize seedlings growth under NaCl stress.” in Line 643 of the manuscript. We then have re-submitted the manuscript.
Thank you for your consideration.
Open Review: I would not like to sign my review report.
Thanks for your positive comments.
Thank you for your consideration.
Quality of English Language: Moderate editing of English language required.
Thanks for your positive comments. As suggested, we have re-edited the whole article in English. We then have re-submitted the manuscript.
Thank you for your consideration.
Does the introduction provide sufficient background and include all relevant references? Must be improved.
Thanks for your positive comments. As suggested, we have carefully improved the English language. We then have re-submitted the manuscript.
Thank you for your consideration.
Is the research design appropriate? Yes.
Thanks for your positive comments.
Thank you for your consideration.
Are the methods adequately described? Must be improved.
T Thanks for your positive comments. As suggested, we have improved the descriptions of method in the manuscript in detail. We then have re-submitted the manuscript.
Thank you for your consideration.
Are the results clearly presented? Can be improved.
Thanks for your positive comments. As suggested, we have carefully revised and improved our results section in the manuscript. We then have re-submitted the manuscript.
Thank you for your consideration.
Are the conclusions supported by the results? Yes.
Thanks for your positive comments. As suggested, we have improved the conclusion section, namely: “In conclusion, 150 μM exogenous MT application could enhance the NaCl resistance of maize seedlings through the following pathways: (1) MT activated photosynthetic pigment biosynthesis genes expression (Zm00001d011819, Zm00001d017766), prevented chlorophyll degradation, and improved stomata morphology, thus improving photosynthetic capacity under NaCl stress. (2) MT up-regulated the expression of antioxidant enzyme genes (Zm00001d009990, Zm00001d047479, Zm00001d014848, Zm00001d007234), which enhanced the activities of antioxidant enzymes, reduced excessive accumulation of ROS, and inhibited membrane lipid peroxidation. (3) These candidate genes, growth phenotypes, and physiological metabolisms interacted with each other to formed a complex regulatory mechanism to response salt tolerance under NaCl stress and MT stimulation. Overall, the application of exogenous MT has the potential to improve maize seedlings growth under NaCl stress.” In Lines 715-728 of the manuscript. We then have re-submitted the manuscript. Thank you for your consideration.
Reviewer 2:
Melatonin, as a novel plant hormone, can effectively regulate plant growth and development under abiotic stress. This manuscript investigated the effects of melatonin on maize seedlings under NaCl stress. Here are some comments:
Thanks for your positive comments.
Thank you for your consideration.
1.The introduction needs to be strengthened by discussing the impact of melatonin on plant under abiotic stress and its mechanisms.
Thanks for your positive comments. As suggested, we have revised the content, namely: “Melatonin (MT), an indole compound, first identified in 1995, and found in both plants and animals, has been revealed to play important roles in growth and development, and stress response of plants [11–13]. Previous studies proved that MT could improve plant resistance against various environmental stresses, including salt, heavy metal, drought, and high/low temperature [14–16]. In particular, foliar spraying of MT and root irrigation had significant alleviation effects in rice (Oryza sativa L.) resistance to salt stress [17]. MT promotes the production of salt-tolerant proteins, antioxidant enzymes, and defense-related molecules, thereby protecting potato (Solanum tuberosum L.) from the harmful effects of abiotic stress [18]. Under salt stress, 70 μM MT treatments increases barley (Hordeum vulgare L.) plants’ photosynthetic efficiency, improves chlorophyll contents, reduces ROS (reactive oxygen species) generation, and thus alleviates oxidative damage to plants [19]. Under 150 mM NaCl stress, there are significant increases appeared in leaf area, biomass, and photosynthesis efficiency of Brassica juncea L. with MT treatments [20]. Thus, exogenous MT application could be approach to alleviate stress injury in multiple plant species [21].” in Lines 69-85 of the manuscript. We then have re-submitted the manuscript.
Thank you for your consideration.
2.The manuscript's English requires improvement, as much of it seems to be machine-translated directly from Chinese.
Thanks for your positive comments. As suggested, we have carefully improved the English language. We then have re-submitted the manuscript.
Thank you for your consideration.
3.It is suggested to supplement data on the relative gene expression changes of POD. As maize is a common crop, the gene sequences of POD have already been identified.
Thanks for your positive comments. As suggested, we very much agree with you that POD-related gene expression data is very important. We have quantited several others in the system. We hope you can consider that due to the urgency of time, we have not made them in a short time. We then have re-submitted the manuscript.
Thank you for your consideration.
- Improve Figure 8. NaCl accumulation is unreasonable, and salt treatment did not inhibit the antioxidant system. Antioxidant enzyme activity and gene expression in different maize genotypes under salt treatment were significantly increased compared to the control (Figure 3 and Figure 7). Additionally, the readability of Figure 8 is poor; increase the font size.
Thanks for your positive comments. As suggested, we redid Figure 8, modified some of it, and increased the font size. We then have re-submitted the manuscript.
Figure 8. Schematic model of response mechanism for exogenous melatonin (MT) alleviates salt injury in maize seedlings. Solid arrows indicated promote effects, dashed arrows indicated inhibited effects; red arrows indicated positive expression of genes; ROS: reactive oxygen species; SOD: superoxide dismutase; POD: peroxidase, CAT: catalase; APX: ascorbate peroxidase; Pn; net photosynthetic rate; Gs: stomatal conductance; Tr: transpiration rate.
Thank you for your consideration.
- There are many careless errors in the manuscript. For example, lines 70-73, capitalization. The full name of ROS, Brassica juncea should be italicized.
Thanks for your positive comments. As suggested, we have revised the content, namely: “MT treatments increases barley (Hordeum vulgare L.) plants’ photosynthetic efficiency, improves chlorophyll contents, reduces ROS (reactive oxygen species) generation, and thus alleviates oxidative damage to plants [19]. Under 150 mM NaCl stress, there are significant increases appeared in leaf area, biomass, and photosynthesis efficiency of Brassica juncea L. with MT treatments [20]. Thus, exogenous MT application could be approach to alleviate stress injury in multiple plant species [21].” in Lines 79-86 of the manuscript. We then have re-submitted the manuscript.
Thank you for your consideration.
- The abbreviations for the four treatments in Figure 3 need modification.
Thanks for your positive comments. As suggested, we changed the abbreviations for the four treatments in Figure 3. We have revised and improved the manuscript. We then have re-submitted the manuscript.
Thank you for your consideration.
Figure 3. Changes of Antioxidant Enzyme Activities and H2O2 Content of all maize seedlings under different treatments. Different lowercase letters indicated significant differences in p < 0.05 level. SOD: catalase; POD: peroxidase; CAT: catalase; APX: ascorbate peroxidase. CK: 0 mM NaCl+0 μM melatonin (MT) treatment; SS: 180 mM NaCl+0 μM MT treatment; M–CK: 0 mM NaCl+150 μM MT treatment; M–SS: 180 mM NaCl+150 μM MT treatment.
- The authors need to carefully check all result analyses. Like lines 130-134, not all genotypes of maize seedlings under NaCl stress showed significantly lower levels of Chl b and Chl a/b compared to the control; in fact, the car content in LX 19 maize seedlings significantly increased. Similarly, there are numerous errors in Section 2.4 as well.
Thanks for your positive comments. As suggested, we made changes throughout the results analysis section. We then have re-submitted the manuscript.
Thank you for your consideration.
- Line 100, what is "more significantly," the authors need to further understand the significance test. If emphasizing related results, it is recommended to conduct significance tests at the 0.01 level.
Thanks for your positive comments. As suggested, we have revised the content, namely: “Under 180 mM NaCl+0 μM MT treatment (SS), the seedling length (SL), seedling leaf area (SLA), seeding fresh weight (SFW), and root fresh weight (RFW) of eight maize genotypes were less than that was treated with 0 mM NaCl+0 μM MT treatment (CK), there were Significant decreases with an average of 23.26%, 33.24%, 37.32, and 28.89%, respectively (Table 1). Indicating that NaCl stress had a negative effect on seedling growth and biomass accumulation in maize. These four traits were increased in 0 mM NaCl+150 μM MT treatment (M–CK) compared to CK by 10.08%, 16.60%, 16.73%, and 18.10%, respectively (Table 1). At the same time, they were increased in 180 mM NaCl+150 μM MT treatment (M–SS) compared to SS treatment, and there were significant increases by 17.01%, 28.61%, 32.32%, and 25.01%, respectively (Table 1). Suggesting that MT application could significantly alleviate salt injury of maize seedlings to maintain overall health and growth potential.” in Lines 102-115 of the manuscript. We then have re-submitted the manuscript.
Thank you for your consideration.
- Check data for Seeding fresh weight and Root fresh weight in Table 1.
Thanks for your positive comments. As suggested, we have carefully checked and changed data for Seeding fresh weight and Root fresh weight in Table 1.
Table 1. Changes of phenotypes observations of all maize seedlings under different treatments
Different lowercase letters indicated significant differences in p < 0.05 level. SL: seedling length; SLA: seedling leaf area; SFW: seedling fresh weight; RFW: root fresh weight. CK: 0 mM NaCl+0 μM melatonin (MT) treatment; SS: 180 mM NaCl+0 μM MT treatment; M–CK: 0 mM NaCl+150 μM MT treatment; M–SS: 180 mM NaCl+150 μM MT treatment.
Thank you for your consideration.
- Line 112, reference citation needed.
Thanks for your positive comments. As suggested, we have revised the content and added the References, namely: “Photosynthetic parameters can reflect the physiological status and changes of maize seedlings under salt stress [41].” in Lines 133-134 of the manuscript. We then have re-submitted the manuscript.
Thank you for your consideration.
- Provide experimental data or references for selected salt and melatonin concentrations.
Thanks for your positive comments. As suggested, we quite agree with you. According to previous work in our team (Chen, F., Fang, P., Peng, Y., Zeng, W., Zhao, X., Ding, Y., Zhuang, Z., Gao, Q. and Ren, B., 2019. Comparative proteomics of salt-tolerant and salt-sensitive maize inbred lines to reveal the molecular mechanism of salt tolerance. International Journal of Molecular Sciences, 20(19), .4725.), they screened out the suitable concentration of NaCl stress in maize was 180 mM NaCl. Therefore, in this study, we used the 180 mM NaCl concentration to perform our study.
Before we start our this experiment, we read a large of reference in recent years, we found that the optimum concentration of exogenous melatonin (MT) was 100~180 μM, that can significant improve various stresses injury (drought, reduces Cd absorption) in different plants (Ahmad, S., Wang, G.Y., Muhammad, I., Farooq, S., Kamran, M., Ahmad, I., Zeeshan, M., Javed, T., Ullah, S., Huang, J.H. and Zhou, X.B., Application of melatonin-mediated modulation of drought tolerance by regulating photosynthetic efficiency, chloroplast ultrastructure, and endogenous hormones in maize. Chemical and Biological Technologies in Agriculture, 2022, 9, 1-14; Xu, L., Xue, X., Yan, Y., Zhao, X., Li, L., Sheng, K. and Zhang, Z. Silicon Combined with Melatonin Reduces Cd Absorption and Translocation in Maize. Plants, 2023,12, 3537.), especially, wang (Wang H. Response of melatonin at different concentrations to loquat seedlings under cold stress. 2020, 5, 102.) reported that 150 μM MT treatment alleviated the damage of loquat seedlings under low temperature stress; Ahmad et al. (Ahmad, S., Guo YW., Ihsan M., Saqib F., Muhammad K., Irshad A., Muhammad Z.; et al. Application of melatonin-mediated modulation of drought tolerance by regulating photosynthetic efficiency, chloroplast ultrastructure, and endogenous hormones in maize. Chemical and Biological Technologies in Agriculture, 2022, 9, 1-14) showed that 150 μM MT application can improved photosynthetic efficiency, chloroplast ultrastructure, and endogenous hormones levels in maize under drought stress. In these regards, in this study, we selected the 150 μM MT concentration to study the alleviation effects in eight maize genotypes seedlings under 180 mM NaCl stress.
Thank you for your consideration.
- Specify the reagent kit number for antioxidant enzyme activity determination.
Thanks for your positive comments. We agree with your views and suggestions. In this study, we used the Solarbio kits (Beijing Solarbio Science and Technology Co., Ltd., Beijing, China) to assay the H2O2 content and four enzymes activities, including superoxide dismutase (SOD), peroxidase (POD), catalase (CAT), and ascorbate peroxidase (APX). Therefore, we have revised the contents in Materials and Methods section, namely: “Refer to the method of Zhao et al. [40], the SOD activity, POD activity, CAT activity, APX activity, and H2O2 content were determined using corresponding Solarbio kits (Beijing Solarbio Science and Technology Co., Ltd., Beijing, China) and using the multi-function microplate reader (SynergyHTX; BioTek Instruments, Inc. USA), following the manufacturer’s kit instructions, respectively. ” in Lines 775-779 of the manuscript. We then have re-submitted the manuscript.
Thank you for your consideration.
- Line 159, SOD activities?
Thanks for your positive comments. As suggested, we have revised the content, namely: “Moreover, M–CK treated leaves of the eight maize varieties, which showed a 20.15% increase with the average SOD activity, a 22.96% increase with average POD activity, a 36.26% increase with average CAT activity, and a 13.87% increase with average APX activity compared to CK treatment (Figure 3).” in Lines 206-209 of the manuscript. We then have re-submitted the manuscript.
Thank you for your consideration.
- Correlation analysis in Figure 6 should include gene expression levels.
Thanks for your positive comments. As suggested, we have revised the content, namely: “To further understand the complex relationship networks of 26 traits and six candidate genes expression, their Pearson correlation analysis was performed across LX19 and NX40-6 leaves under four treatments (Figure 7). Interestingly, there were 207 groups with significant (p < 0.01 or p < 0.05) correlations between both traits, 15 groups with significant (p < 0.01 or p < 0.05) correlations between both genes, as well as 119 groups with significant (p < 0.01 or p < 0.05) correlations between trait and gene (Figure 7). Such as, the expression levels of Zm00001d047479 (Cu-Zn SOD) showed significantly positive correlation to REC, Ci, SOD activity, POD activity, CAT activity, APX activity, and H2O2 level, and showed significantly negative correlation to SL, SLA, SFW, RFW, RWC, Pn, Gs, Tr, SL–F, SW–F, SA–F, SW–Z, Chl a content, Chl b content, Chl a/b, and SPAD value. The expression levels of photosynthetic pigments biosynthesis related gene, i.e., Zm00001d017766, showed significantly positive correlation to SL, RFW, Pn, Gs, Tr, SW–F, SA–F, SW–Z, Chl a content, Chl b content, Car content, Chla/b, and SPAD value, and showed significantly negative correlation to REC, Ci, SOD activity, CAT activity, APX activity, and H2O2 content (Figure 7). These findings thus showed that the six candidate genes for antioxidant enzyme activity and photosynthetic pigment biosynthesis could cooperate with each other to directly or indirectly regulate multiple metabolism and development processes, including seedling growth, stomatal morphology, photosynthetic performance, antioxidant system, ROS homeostasis, and membrane integrity in maize under diverse environments.” in Lines 346-365 of the manuscript. We then have re-submitted the manuscript.
Thank you for your consideration.
Figure 7. Pearson correlation analysis among 26 traits and six candidate genes across LX19 and NX40–6 seedlings under four treatments. SL: seedling length; SLA: seedling leaf area; SFW: seedling fresh weight; RFW: root fresh weigh; REC: relative conductivity; RWC: relative water content; Pn: net photosynthetic rate; Gs: stomatal conductance; Ci: intercellular CO2 concentration; Tr; transpiration rate; SL–Z: stomatal length of adaxial surface; SW–Z: stomatal width of adaxial surface; SA–Z: stomatal area of adaxial surface; SL–F: stomatal length of abaxial surface; SW-F: stomatal width of abaxial surface; SA–F: stomatal area of abaxial surface; Chl a: chlorophyll a content; Chl b: chlorophyll b content; Car: carotenoid content; Chl a/b: chlorophyll a content/chlorophyll b content; SPAD: SPAD content; SOD: superoxide dismutase activity; POD: peroxidase activity; CAT: catalase activity; APX: ascorbate peroxidase activity; H2O2: H2O2 content. ** indicated significant correlations in p < 0.01 level. while * indicated significant correlations in p < 0.05 level.
Thank you for your consideration
- Line 330, 150 um?
Thanks for your positive comments. As suggested, we changed “150 um” to “150 μM”, namely: “Consistently, our research showed that salt stress caused 40.98% increase in H2O2 accumulation in maize leaves, but 150 μM exogenous MT obviously alleviated H2O2 accumulation (Figure 3E).” in Lines 482-484 of the manuscript. We then have re-submitted the manuscript.
Thank you for your consideration.
- The sentences in the manuscript should use passive voice and past tense, such as in Sections 2.4, 4.4, and 4.5, etc.
Thanks for your positive comments. As suggested, we have adjusted the tenses of the sentences in the manuscript. We then have re-submitted the manuscript.
Thank you for your consideration
- Line 449, confirm whether standard deviation or standard error is used.
Thanks for your positive comments. As suggested, all data are shown as means ± SE (standard error). We then have re-submitted the manuscript.
Thank you for your consideration.
- Line 459, were the gene sequences used in the manuscript designed by authors or derived from references (38 and 39)? The enzyme gene is incorrect.
Thanks for your positive comments. We strongly agree with you that the genes used in the manuscript were designed by the author. As suggested, we have revised the content, namely: “According to the changes of phenotypes and physiological metabolisms of eight maize genotypes under all treatments, we selected LX19 and NX40–6 seedlings, with contrasting salt tolerance to extract their total RNAs with TRIZOL reagent (Invitrogen, USA). Which was then reverse–transcribed into cDNA using a SuperScript III First strand Kit (Invitrogen). The qRT-PCR was conducted using TransStart Tip Green qPCR SuperMix (Tran, Beijing, China). Primers (Table S1) for these candidate genes for antioxidant enzymes and photosynthetic pigment biosynthesis were designed via the Primer3web v.4.1.0 (https://primer3.ut.ee/; accessed on 1 May 2024). The positions of these genes were mapped in the Zea_mays B73_V4 reference genome (https://www.maizegdb.org/; accessed on 12 May 2024), and their functional annotation was performed using the tool AgBase v.2.00 (https://agbase.arizona.edu/; accessed on 18 May 2024). Relative gene expression levels were calculated by the 2−∆∆Ct method, with Zm00001d010159 as an internal reference gene.” in Lines 644-658 of the manuscript. We then have re-submitted the manuscript.
Thank you for your consideration.
- When using the 2−∆∆Ctmethod for relative gene expression calculation, the relative expression level of the control should be around 1 (2-0); confirm Figure 7.
Thanks for your positive comments. As suggested, we agree with you very much and we have carefully examined our data and redone Figure 7. We then have re-submitted the manuscript.
Thank you for your consideration.
Figure 6. Relative expression levels of six random candidate genes across LX19 and NX40-6 maize varieties under four treatment conditions. The different letter indicated significant differences between two treatments within a maize genotype (p < 0.05). Different lowercase letters indicated significant differences in p < 0.05 level. CK: 0 mM NaCl+0 μM melatonin (MT) treatment; SS: 180 mM NaCl+0 μM MT treatment; M–CK: 0 mM NaCl+150 μM MT treatment; M–SS: 180 mM NaCl+150 μM MT treatment.
- The conclusion does not address the relevant results of gene expression.
Thanks for your positive comments. As suggested, we have improved the conclusion section, namely: “In conclusion, 150 μM exogenous MT application could enhance the NaCl resistance of maize seedlings through the following pathways: (1) MT activated photosynthetic pigment biosynthesis genes expression (Zm00001d011819, Zm00001d017766), prevented chlorophyll degradation, and improved stomata morphology, thus improving photosynthetic capacity under NaCl stress. (2) MT up-regulated the expression of antioxidant enzyme genes (Zm00001d009990, Zm00001d047479, Zm00001d014848, Zm00001d007234), which enhanced the activities of antioxidant enzymes, reduced excessive accumulation of ROS, and inhibited membrane lipid peroxidation. (3) These candidate genes, growth phenotypes, and physiological metabolisms interacted with each other to formed a complex regulatory mechanism to response salt tolerance under NaCl stress and MT stimulation. Overall, the application of exogenous MT has the potential to improve maize seedlings growth under NaCl stress.” In Lines 715-728 of the manuscript. We then have re-submitted the manuscript.
Thank you for your consideration.
- Reference format needs to be consistent, such as journal name abbreviations and page numbers.
Thanks for your positive comments. As suggested, we have carefully checked and modified the all references. We then have re-submitted the manuscript.
Thank you for your consideration
Open Review: I would not like to sign my review report.
Thanks for your positive comments.
Thank you for your consideration.
Quality of English Language: Extensive editing of English language required.
Thanks for your positive comments. As suggested, we have carefully improved the English language. We then have re-submitted the manuscript.
Thank you for your consideration.
Does the introduction provide sufficient background and include all relevant references? Can be improved.
Thanks for your positive comments. As suggested, we have revised and improved the introduction section. We then have re-submitted the manuscript.
Thank you for your consideration.
Is the research design appropriate? Can be improved.
Thanks for your positive comments. As suggested, we have improved the experiment design in the manuscript. We then have re-submitted the manuscript.
Thank you for your consideration.
Are the methods adequately described? Must be improved.
Thanks for your positive comments. As suggested, we have improved the descriptions of method in the manuscript in detail. We then have re-submitted the manuscript.
Thank you for your consideration.
Are the results clearly presented? Must be improved.
Thanks for your positive comments. As suggested, we have carefully revised and improved our results section in the manuscript. We then have re-submitted the manuscript.
Thank you for your consideration.
Are the conclusions supported by the results? Can be improved.
Thanks for your positive comments. As suggested, we have improved the conclusion section, namely: “In conclusion, 150 μM exogenous MT application could enhance the NaCl resistance of maize seedlings through the following pathways: (1) MT activated photosynthetic pigment biosynthesis genes expression (Zm00001d011819, Zm00001d017766), prevented chlorophyll degradation, and improved stomata morphology, thus improving photosynthetic capacity under NaCl stress. (2) MT up-regulated the expression of antioxidant enzyme genes (Zm00001d009990, Zm00001d047479, Zm00001d014848, Zm00001d007234), which enhanced the activities of antioxidant enzymes, reduced excessive accumulation of ROS, and inhibited membrane lipid peroxidation. (3) These candidate genes, growth phenotypes, and physiological metabolisms interacted with each other to formed a complex regulatory mechanism to response salt tolerance under NaCl stress and MT stimulation. Overall, the application of exogenous MT has the potential to improve maize seedlings growth under NaCl stress.” In Lines 715-728 of the manuscript. We then have re-submitted the manuscript. Thank you for your consideration.
Sincerely,
Xiaoqiang Zhao professor
State Key Laboratory of Aridland Crop Science, Gansu Agricultural University
E-mail: zhaoxq3324@163.com

Reviewer 2 Report
Comments and Suggestions for Authors
Melatonin, as a novel plant hormone, can effectively regulate plant growth and development under abiotic stress. This manuscript investigated the effects of melatonin on maize seedlings under NaCl stress. Here are some comments:
1. The introduction needs to be strengthened by discussing the impact of melatonin on plant under abiotic stress and its mechanisms.
2. The manuscript's English requires improvement, as much of it seems to be machine-translated directly from Chinese.
3. It is suggested to supplement data on the relative gene expression changes of POD. As maize is a common crop, the gene sequences of POD have already been identified.
4. Improve Figure 8. NaCl accumulation is unreasonable, and salt treatment did not inhibit the antioxidant system. Antioxidant enzyme activity and gene expression in different maize genotypes under salt treatment were significantly increased compared to the control (Figure 3 and Figure 7). Additionally, the readability of Figure 8 is poor; increase the font size.
5. There are many careless errors in the manuscript. For example, lines 70-73, capitalization. The full name of ROS, Brassica juncea should be italicized.
6. The abbreviations for the four treatments in Figure 3 need modification.
7. The authors need to carefully check all result analyses. Like lines 130-134, not all genotypes of maize seedlings under NaCl stress showed significantly lower levels of Chl b and Chl a/b compared to the control; in fact, the car content in LX 19 maize seedlings significantly increased. Similarly, there are numerous errors in Section 2.4 as well.
8. Line 100, what is "more significantly," the authors need to further understand the significance test. If emphasizing related results, it is recommended to conduct significance tests at the 0.01 level.
9. Check data for Seeding fresh weight and Root fresh weight in Table 1.
10. Line 112, reference citation needed.
11. Provide experimental data or references for selected salt and melatonin concentrations.
12. Specify the reagent kit number for antioxidant enzyme activity determination.
13. Line 159, SOD activities?
14. Correlation analysis in Figure 6 should include gene expression levels.
15. Line 330, 150 um?
16. The sentences in the manuscript should use passive voice and past tense, such as in Sections 2.4, 4.4, and 4.5, etc.
17. Line 449, confirm whether standard deviation or standard error is used.
18. Line 459, were the gene sequences used in the manuscript designed by authors or derived from references (38 and 39)? The enzyme gene is incorrect.
19. When using the 2−∆∆Ct method for relative gene expression calculation, the relative expression level of the control should be around 1 (2-0); confirm Figure 7.
20. The conclusion does not address the relevant results of gene expression.
21. Reference format needs to be consistent, such as journal name abbreviations and page numbers.
Comments on the Quality of English LanguageExtensive editing of English language required.
Author Response

(The authors gave the same response as above.)

Round 2
Reviewer 1 Report
Comments and Suggestions for Authors
Reviewer 1
The manuscript entitled “Exogenous Melatonin Alleviates NaCl Injury by Influencing
Stomatal Morphology, Photosynthetic Performance and Antioxidant Balance in Maize” has been improved, and the authors have addressed most of my reviews. However, there are minor issues to consider.
The introduction needs to be expanded. Consider adding more details using recent studies on the different pathways that crop in general and maize in your case developed to overcome salt stress. Also talk about the effect of salt stress on plants’ photosynthesis or physiology.
Lines 22 to 25: Start a new sentence by writing this: [Meanwhile, we examined the expression levels of six antioxidant enzyme activities- and photosynthetic pigments biosynthesis-related genes, to … maize]
Lines 25 to 30: You did not consider rewriting understandable and concise sentences.
Write the following: [The results showed … under salt stress. The relative conductivity and H2O2 level “were” reduced by … compared to only salt-treated seedlings.
Line 30 to 34: Rewrite as follows: [The improvement of … seedlings “were” treated … salt stress”;” their stomatal … respectively.] Delete “averaged” in the sentence.
Line 34: Do not capitalize “Gene”
Line 35: […antioxidant enzyme-“related” genes]
Line 45 and 46: I suggest you rephrase the sentence: “Soil salinity is one of the abiotic stresses that severely limits agricultural production across the globe.”
Lines 58 to 60: Please rewrite this part. It is still confusing.
Lines 66 and 67: Please provide some of the chemicals or plant growth regulators and cite the sources.
Line83: Delete “appeared”
Line 85: could be “an” approach
Line 89: Delete “on”
Lines 96 to 99: Start with [“Our findings provide a valuable… process.”]
Go through the manuscript and delete the word “average.” When you talk about enhancement and reduction of any parameters, it is clear to the readers that you are talking about means after statistical analyses. It is annoying while reading.
Lines 201 to 206: In this single sentence, you used the word “significantly” three times. Rewrite the sentence. Your writing is not fluid. This is one of the reasons I recommend you write short sentences. You can rephrase this way “SOD activity in the leaves of all maize seedlings subjected to SS treatment increased by 16.96% to 82.03% compared to CK, except GS36–2 seedlings. POD activity increased by 18.30% to 106.00%, CAT activity by 48.97% to 163%, and APX activity by 17.86% to 42.75% (Figure 3), all reflecting significant increases.”
Lines 244 to 248: Like the previous comment, you used “clearly” redundantly. Rephrase the sentence.
Go through the whole manuscript to correct wherever you used repetitive expressions in single sentences. Redundancy does not make your writing fluid, making it difficult for readers to get the message behind your findings.
You did a great job designing Figure 8, which is clear enough.
Conclusion: You can start saying: [ In conclusion, … enhance maize seedlings “tolerance” to NaCl stress through …]
Comments on the Quality of English Language
Carefully check the grammar and the redundancy of expressions through out the manuscript.
Reviewer 2 Report
Comments and Suggestions for Authors
Accept in present form.